

# Malware detection using static analysis in Android: a review of FeCO (features, classification, and obfuscation)

Rosmalissa Jusoh[1], Ahmad Firdaus[1], Shahid Anwar[2], Mohd Zamri Osman[1], Mohd Faaizie Darmawan[3] and Mohd Faizal Ab Razak[1]

[1] Faculty of Computing, College of Computing and Applied Sciences, Universiti Malaysia Pahang, Pekan, Pahang, Malaysia
[2] Department of Information Engineering Technology, National Skills University, Islamabad, Pakistan
[3] Faculty of Computer & Mathematical Sciences, Universiti Teknologi Mara, Tapah, Perak, Malaysia

## ABSTRACT

Android is a free open-source operating system (OS), which allows an in-depth understanding of its architecture. Therefore, many manufacturers are utilizing this OS to produce mobile devices (smartphones, smartwatch, and smart glasses) in different brands, including Google Pixel, Motorola, Samsung, and Sony. Notably, the employment of OS leads to a rapid increase in the number of Android users. However, unethical authors tend to develop malware in the devices for wealth, fame, or private purposes. Although practitioners conduct intrusion detection analyses, such as static analysis, there is an inadequate number of review articles discussing the research efforts on this type of analysis. Therefore, this study discusses the articles published from 2009 until 2019 and analyses the steps in the static analysis (reverse engineer, features, and classification) with taxonomy. Following that, the research issue in static analysis is also highlighted. Overall, this study serves as the guidance for novice security practitioners and expert researchers in the proposal of novel research to detect malware through static analysis.

## INTRODUCTION

Mobile devices, such as smartphones, iPads, and computer tablets, have become everyday necessities to perform important tasks, including education, paying bills online, bank transactions, job information, and leisure. Based on the information from an online mobile device production website, Android is one of the popular operating systems (OS) used by manufacturers (*Rayner, 2019*; *Jkielty, 2019*). The open-source platform in Android has facilitated the smartphone manufacturers in producing Android devices of various sizes and types, such as smartphones, smartwatches, smart televisions, and smart glasses. In the most recent decades, the quantity of remarkable Android gadgets accessible worldwide has increased from 38 in 2009 to over 20,000 in 2016 (*Android, 2019a*). As a result of the demand for this Android OS, the recent statistics from Statista revealed that the number of Android malware increase to 26.6 million in March 2018 (*Statista, 2019*). Moreover,

Corresponding authors
Rosmalissa Jusoh,
rosmalissa@ump.edu.my
Ahmad Firdaus,
firdausza@ump.edu.my

McAfee discovered a malware known as Grabos, which compromises the Android and breaches Google Play Store security (*McAfee, 2019*). It was also predicted that 17.5 million Android smartphones had downloaded this Grabos mobile malware before they were taken down.

Mobile malware is designed to disable a mobile device, allow malicious acts to remotely control the device, or steal personal information (*Beal, 2013*). Moreover, these malicious acts able to run stealthily and bypass permission if the Android kernel is compromised by mobile malware (*Ma & Sharbaf, 2013*; *Aubrey-Derrick Schmidt et al., 2009b*). In September 2019, a total of 172 malicious applications were detected on Google Play Store, with approximately 330 million installations. According to researchers, the malicious components were hidden inside the functional applications. When the applications are downloaded, it leads to the appearance of popup advertisements, which remain appear even when the application was closed (*O'Donnell, 2019*). To detect this malware, security practitioners conducting malware analysis, which aims to study the malware characteristics and behaviour. There are dynamic, static, and hybrid analysis.

Table 1 shows comparison for static, dynamic and hybrid analysis done from previous researches. Specifically, dynamic analysis is an analysis, which studies the execution and behaviour of the malware (*Enck, 2011*; *Yaqoob et al., 2019*). However, dynamic analysis is incapable of identifying several parts of the code operating outside the monitoring range. Besides, provided that the dynamic analysis is a high resource-consuming analysis with a high specification for hardware (*Enck, 2011*), static analysis is another alternative to detect malware. It is an analysis, which examines malware without executing or running the application. Additionally, this analysis able to identify malware more accurately, which would act under unusual conditions (*Castillo, 2011*). This is due to static analysis examine overall parts of a program including parts that excluded in dynamic analysis. Furthermore, static analysis is able to detect unknown malware just as dynamic analysis could (*Yerima, Sezer & McWilliams, 2014*) and requiring low resources.

To integrate the characteristics of the static and dynamic method, three-layer detection model called SAMAdroid has been proposed by *Saba Arshad et al. (2018)* which combines static and dynamic characteristics. Mobile Sandbox by *Spreitzenbarth et al. (2015)* which proposed to use the results of static analysis to guide the dynamic analysis and finally realize classification. The hybrid analysis technique is great to help in improving the accuracy, but it also has a major drawback such as the waste of time and space for the huge number of malware samples to be detected and analyzed (*Fang et al., 2020*; *Alswaina & Elleithy, 2020*).

Table 2 presents the past review articles on Android, with *Feizollah et al. (2015)* specifically focusing on features, including static, dynamic, hybrid, and metadata. It summarizes the features preferred researchers in their analysis. Comparatively, this study placed more emphasis on features besides classification and obfuscation. Subsequent reviews, namely *Sufatrio et al. (2015)* and *Schmeelk, Yang & Aho (2015)*, highlighted the survey, taxonomy, challenges, advantages, limitations in the existing research in the Android security area, and the technique in the static analysis research on Android. However, compared to the current review, the aforementioned reviews only presented a few features and information on static analysis. In the Android permission category (*Fang, Han & Li,*

**Table 1  Comparison malware analysis techniques.** Previous works compared using static, dynamic, and hybrid techniques.

| Year | References | Analysis | Features |
|------|-----------|----------|----------|
| 2020 | *Fang et al. (2020)* | Static | Texture, color, text |
| 2019 | *Qiu et al. (2019)* | Static | permissions, API calls, network addresses |
| 2019 | *Zhang, Thing & Cheng (2019)* | Static | Assembly, Dex, Xml, Apk |
| 2019 | *Xu, Ren & Song (2019)* | Static | CFG, DFG |
| 2019 | *Omid Mirzaeiq et al. (2019)* | Static | API calls |
| 2019 | *Vega & Quintián (0000)* | Static | Repackaging and standalone |
| 2019 | *Vega et al. (2019)* | Static | Root node, decision nodes, and leaf nodes |
| 2019 | *Fasano et al. (2019)* | Static | |
| 2019 | *Blanc et al. (2019)* | Static | Code metric |
| 2019 | *Xie et al. (2019)* | Static | Platform-based permissions, hard- ware components, and suspicious API calls |
| 2019 | *Turker & Can (2019)* | Static | Permissions and API calls |
| 2018 | *Atzeni et al. (2018)* | Hybrid | Manifest file (i.e., number of activities, permissions, receivers, filters), and the source code analysis |
| 2018 | *Kim et al. (2019)* | Hybrid | API call |
| 2018 | *Ming Fan et al. (2018)* | Static | Weighted-sensitive-API-call-based graph |
| 2018 | *Sun et al. (2018, 2019)* | Dynamic | Enabling the recording of parameters and return value of an API call |
| 2018 | *Martín, Rodríguez-Fernández & Camacho (2018)* | Dynamic | transitions probabilities, states frequencies, and aggregated state frequencies grouped |
| 2018 | *Aktas & Sen (2018)* | Hybrid | number of activities, services and receivers given in the Manifest file and the size of the APK file |
| 2018 | *Garcia, Hammad & Malek (2018)* | Static | API usage, reflection-based features, and features from native binaries of apps |
| 2018 | *Calleja et al. (2018)* | Static | API calls, intent actions and information flow |
| 2018 | *Alswaina & Elleithy (2018)* | Static | App's permissions |
| 2017 | *Massarelli et al. (2017)* | Dynamic | Fingerprint |
| 2017 | *Zhou et al. (2017)* | Static | API call graphs |
| 2017 | *Chakraborty, Pierazzi & Subrahmanian (2020)* | Hybrid | API calls, code, Android Manifest, encryption or reflection |
| 2017 | *Sedano et al. (2017b)* | Static | Minimum-Redundancy Maximum- Relevance (MRMR) |
| 2016 | *Battista et al. (2016)* | Static | Java Bytecode |
| 2016 | *Hsiao, Sun & Chen (2016)* | Dynamic | API call |
| 2016 | *González, Herrero & Corchado (2017)* | Static | API call and the names of functions and methods |
| 2016 | *Ming Fan et al. (2018)* | Static | Subgraph |
| 2016 | *Kang et al. (2016)* | Static | n-opcode feature |
| 2016 | *Malik & Khatter (2016)* | Dynamic | System call |
| 2016 | *Sedano et al. (2017a)* | Static | Manifest file, apk file |
| 2016 | *Feng et al. (2017)* | Hybrid | Malware signatures |
| 2015 | *Lee, Lee & Lee (2015)* | Static | Signature extraction signature matching |
| 2015 | *Aresu et al. (2016)* | Dynamic | Fine-grained HTTP structural |
| 2015 | *Li et al. (2015)* | Static | API data dependency |
| 2014 | *Deshotels, Notani & Lakhotia (2014a)* | Static | API call, apk |
| 2014 | *Kang et al. (2013)* | Static | Bytecode frequency |
| 2014 | *Suarez-Tangil et al. (2014)* | Static | Code structures |

*2014*), this article reviewed the issues and existed countermeasures for permission-based Android security. It also incorporated the necessary information related to Android permission, namely permission documentation, over-claim of permission, permission administration, and more.

Notably, our review article featured more aspects besides permission by putting an in-depth focus on static analysis. Meanwhile, the other paper, evaluates three anti-viruses, namely Stowaway, AASandbox, and Droidbox. To assist in the decision regarding the better anti-virus, the aforementioned anti-viruses were separated using static and dynamic analyses, followed by a comparison between one another (*Ma & Sharbaf, 2013*). Another related review (*Pan et al., 2020*), provided the common features only and mentioned less the deep learning part in static analysis. However, the information gained from past research, specifically on features, deep learning and obfuscation was still lacking. In comparison to the previous survey articles listed in the Table 2, the main contributions of the current article are as follows:

(a) Present the information of static analysis in detail, specifically the dataset, reverse engineer, features, and classification from 2009 until 2019.

(b) Review on the latest features in ten categories, namely advertisements libraries, application programming interface (API), apk, dex and xml properties, directory path, commands, a function call, geographic location, manifest file, network address, and codebase.

(c) Review the classification section, which includes machine learning, deep learning, graph, and other methods.

(d) Discuss on open research issues, which include the trends in the static analysis, obfuscation, and the list of all previous static analysis experiments.

The remaining section of this article begins with section two, which presents the methodology of this study into four steps. Section three reviews the existing research on static analysis, which concludes the dataset, reverse engineer, features, and classification. Section four discusses the open research issues in the static analysis, followed by section five, which concludes the review.

## SURVEY METHODOLOGY

### Methodology

This section describes the method to retrieve the articles related to malware detection using static analysis for Android. We used Web of Science to run the review, eligibility and exclusion criteria, steps of the review process (identification, screening, eligibility), and data analysis.

### *Identification*

The review was performed based on the main journal database in the Web of Science (WoS). This database covers more than 256 disciplines with millions of journals regarding the subjects related to network security, computer system, development, and planning. It also stores over 100 years of comprehensive backfile and citation data established by Clarivate Analytics (CA), which are ranked through three separate measures, namely

**Table 2** **Comparison with previous review articles.** Summarization of previous related review articles in detecting malware.

| References | Ma & Sharbaf (2013) | Fang, Han & Li (2014) | Feizollah et al. (2015) | Sufatrio et al. (2015) | Schmeelk, Yang & Aho (2015) | Pan et al. (2020) | This paper |
|---|---|---|---|---|---|---|---|
| Title | Investigation of Static and Dynamic Android Anti-virus Strategies | Permission-based Android Security: Issues and Countermeasures | A Review on Feature Selection in Mobile Malware Detection | Securing Android: A Survey, Taxonomy, and Challenges | Android Malware Static Analysis Techniques | A Systematic Literature Review of Android Malware Detection Using Static Analysis | Malware Detection using Static Analysis for Android: A Review and Open Research Issue |
| Year | 2013 | 2014 | 2015 | 2015 | 2015 | 2020 | Current paper |
| Citations | 9 | 132 | 172 | 146 | 21 | 1 | |
| Dataset | | | ✓ | | | ✓ | ✓ |
| Reverse engineer tools | | ✓ | | | | ✓ | ✓ |
| All static features | | | ✓ | ✓ | | ✓ | ✓ |
| All classifications (Machine learning, deep learning, graph, and others) | | | | | | | ✓ |
| Obfuscation constraints and methods to overcome it | | | | | | | ✓ |

citations, papers, and citations per paper. The search strings in the CA database were "static analysis", "malware", and "Android".

There were 430 records identified through database searching. These journals and conferences are mainly from Computer and Security and IEEE Access, which are listed in Table 3. Collections of the studies that are related to Android malware detection using static analysis in the reference section, where studies take up a small proportion in the primary studies. All the studies related to search terms are taken into account, and the searching range is from January 2009 to December 2019.

### Screening

Experiment articles were identified in the static analysis, omitting other unrelated articles. Initially, the searching of articles was specified into a journal article and excluded review articles, books, and conference proceedings. To focus specifically on static analysis, the articles, which combined both static and dynamic analyses, were removed. Another criterion focused on the selection of the articles was the use of English, which therefore removed all non-English articles to avoid any difficulty in translating in the future. The selection of articles took place from 2009 to 2019, totaling the duration to 10 years. This duration was suitable for exploring the evolution of research in security areas. Apart from that, the Android platform was the focus of this study.

### Eligibility

Figure 1 depicts the review that process involved four steps; identification, screening, eligibility, and analysis. The review was performed in mid of 2019. Based on previous studies, the process used similar keywords related to malware detection, static analysis, and security. After the identification process, we remove any duplicated articles. During the screening process, we discover 375 documents and remove a few articles and left 172 articles. This is because the articles were unrelated to the interested area. Lastly, we used 150 articles for review (*Shaffril, Krauss & Samsuddin, 2018*).

### Data analysis included

Then we analyzed the remaining articles, extracted the abstract, and downloaded the full articles. This is to find the appropriate topic in-depth and to have a strong justification for the research. Then, this process organized the topic and subtopic accordingly based on the static analysis. Qualitative analysis was performed based on content analysis to identify issues related to this study.

## Static analysis

Mobile malware compromises Android devices (smartphone, smartwatch, and smart television) for wealth, stealing data, and personal purposes. The examples of mobile malware include root exploit, botnets, worms, and Trojan. To detect malware, most of security practitioners perform two types of analysis; dynamic and static.

Specifically, dynamic analysis is an experiment, which detects malware by executing malware and benign applications to monitor and differentiate their behaviours. However, the monitoring of all behaviours is costly and requires high specifications in terms of device

**Table 3 The main journals and conferences.** Collections of the studies that are related to Android malware detection using static analysis, where studies take up a small proportion in the primary studies. The highest number of journals and conferences are manually counted.

| Category | Acronym | Full name |
| --- | --- | --- |
| Journal | – | IEEE Access |
| | – | Computers and Security |
| | PICECE | Palestinian International Conference on Electrical and Computer Engineering |
| | – | Computer Virology |
| | – | |
| | ASME | Manufacturing Science and Engineering |
| | KSII | Internet and Information Systems |
| | – | |
| | – | Current Bioinformatics |
| | – | Neural Computation |
| | – | Frontiers of Information Technology and Electronic Engineering |
| | – | Neurocomputing |
| | JISA | Information Security and Applications |
| | – | Advances in Intelligent Systems and Computing |
| | – | IEEE Transactions on Information Forensics and Security |
| Conference | ACM | Conference on Multimedia |
| | DSC | IEEE International Conference on Data Science in Cyberspace |
| | FSE | ACM SIGSOFT International Symposium on Foundations of Software Engineering |
| | TrustCom | IEEE International Conference on Trust, Security and Privacy in Computing and Communications |
| | GLOBECOM | IEEE Global Communications Conference |
| | SIN | International Conference on Security of Information and Networks |
| | MALWARE | International Conference on Malicious and Unwanted Software |
| | ICC | IEEE International Conference on Communications |
| | ICISSP | International Conference on Information Systems Security and Privacy |
| | PIMRC | IEEE International Symposium on Personal, Indoor and Mobile Radio Communications |
| | IJCNN | International Joint Conference on Neural Networks |
| | Big Data | IEEE International Conference on Big Data |

memory, CPU, and storage. Furthermore, the malware is inflicted on a device at a certain time or whenever the attacker decides on it. Accordingly, as the dynamic analysis only monitors behaviours at a certain range of time based on the research period, numerous malware activities outside the research period might be omitted (*Feizollah et al., 2013*; *Yerima, Sezer & Muttik, 2015*; *Wei et al., 2017*). Furthermore, dynamic analysis requires a separate and closed virtual environment to run a malware and observe its behaviour on

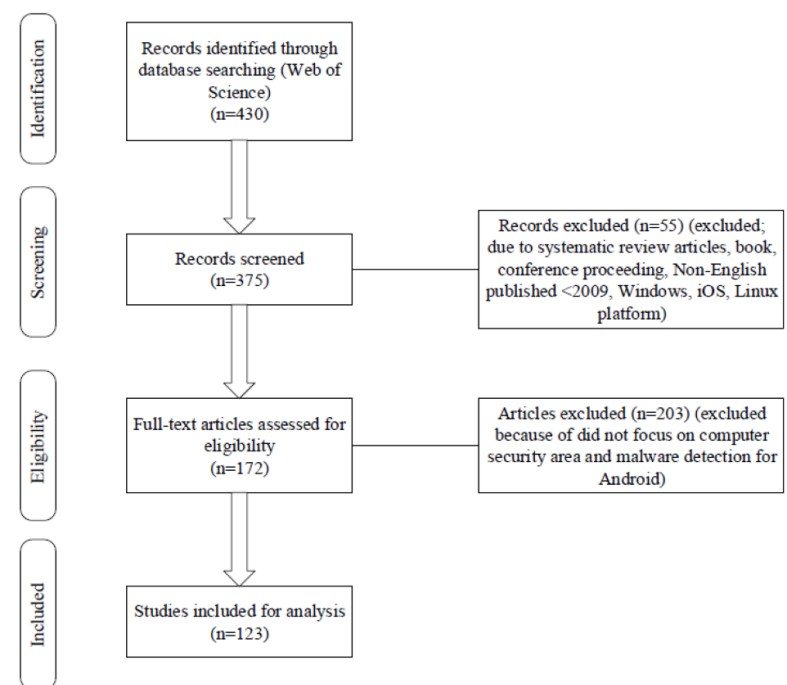

**Figure 1** **The flow diagram of the study.** Each step indicates the number of articles taken from Web of Science based on identification, screening, eligibility, and analysis processes.

the system. However, an isolated setup dynamic leads to an impractical analysis in the Android platform due to the increase in power and memory consumption. While power and memory are the most concerning constraints of Android devices, static analysis is the alternative for the dynamic analysis.

Static analysis is a category of analysis, which investigates the malware application code and examine full activities in an application within an unlimited range of time, by without executing the application (*Chess & McGraw, 2004*). The main step of static analysis procedure is the reverse engineer process, which retrieves the whole code and further scrutinises the structure and substance within the application (*Sharif et al., 2008*; *Chang & Hwang, 2007*; *Aafer, Du & Yin, 2013*). Therefore, this analysis can examine the overall code with low requirement for memory resources and minimum CPU processes. Additionally, the analysis process is prompt due to the absence of the application. With this analysis, unknown malware is also identified using enhanced detection accuracy through machine learning approaches (*Narudin et al., 2014*; *Feizollah et al., 2013*). Table 4 presents the advantages and disadvantages of dynamic and static analyses.

A lot of researchers publish their works using static approaches for malware detection on the Android platform. Even in this static approach, in its turn, contains a number of approaches. For example, there are signature-based approach and other approach are depending on detection and classification of the source code. Signature-based detection utilizes its specification by having an information of malware signatures determined and arranged in advance inspection (*Samra et al., 2019*). However, signature-based approach

**Table 4  Advantages and disadvantages between dynamic, static and hybrid.** Dynamic, static, and hybrid analysis techniques have their own pros and cons. This table summarizes the advantages and disadvantages of these techniques.

| Dynamic | Static | Hybrid |
|---|---|---|
| Advantages | | |
| Able to detect unknown malware | Able to detect unknown malware with the aid of machine learning | Able to detect unknown malware with combination of static and dynamic analysis |
| Able to detect benign applications, which abruptly transform into malware during its execution | The application of reverse engineer takes a short amount of time | |
| | The examination on the overall code, followed by the identification of a possible action | |
| | Low resources (e.g., CPU, memory, network, and storage). Therefore, this analysis is suitable for mobile device which equipped with low specifications. | |
| Limitations | | |
| High resources (e.g., CPU, memory, network, and storage) | Inability to detect normal application, which promptly transforms the malware | Waste of time |
| Higher time consumption to run the application for further analysis and exploration | Obfuscation | Require more spaces for huge number of malware samples |
| Possibly omits the malware activities outside the analysis range | The investigation is continued to determine the minimal features (e.g., permission, a function call, and strings) to detect malware | |
| Difficulty in detecting applications, which can hide malicious behaviour when it is operated | | |
| The investigation is continued to determine the minimal features (e.g., traffic and memory) to detect malware | | |

are not able to detect unknown malware even though this approach is a set of features that uniquely differentiate the executable code (*Gibert, Mateu & Planes, 2020*).

Obfuscation is one of the obstacles in the static analysis, which is used by malware authors in their malicious software to evade the intrusion detection or antivirus system (*Wei et al., 2017*). The examples of the obfuscation methods are renaming the code, adding unnecessary codes, and encrypting the string. Therefore, security practitioners need to overcome obfuscation to increase their detection results. Accordingly, the alternatives performed by the security practitioners are presented in 'Obfuscation'.

Table 4 shows that both static and dynamic analyses have similar limitations despite the selection of the ideal features in minimal amount. In detecting malware, features refer to the attributes or elements to differentiate an application, which may either be malware or benign. Security practitioners are faced with obstacles in investigating various features in all types of categories (e.g., permission, API, directory path, and code-based) and the need to simultaneously reduce these features. Notably, determining the ideal features in minimal amount is crucial to enhance the accuracy of the analyses (e.g., the accuracy of the predictive model) and reduce data and model complexity (*Feizollah et al., 2015*).

Figure 2 illustrates the static analysis operation, which consisted of several steps. The first step was the acquirement of the benign and malware datasets in the Android application,

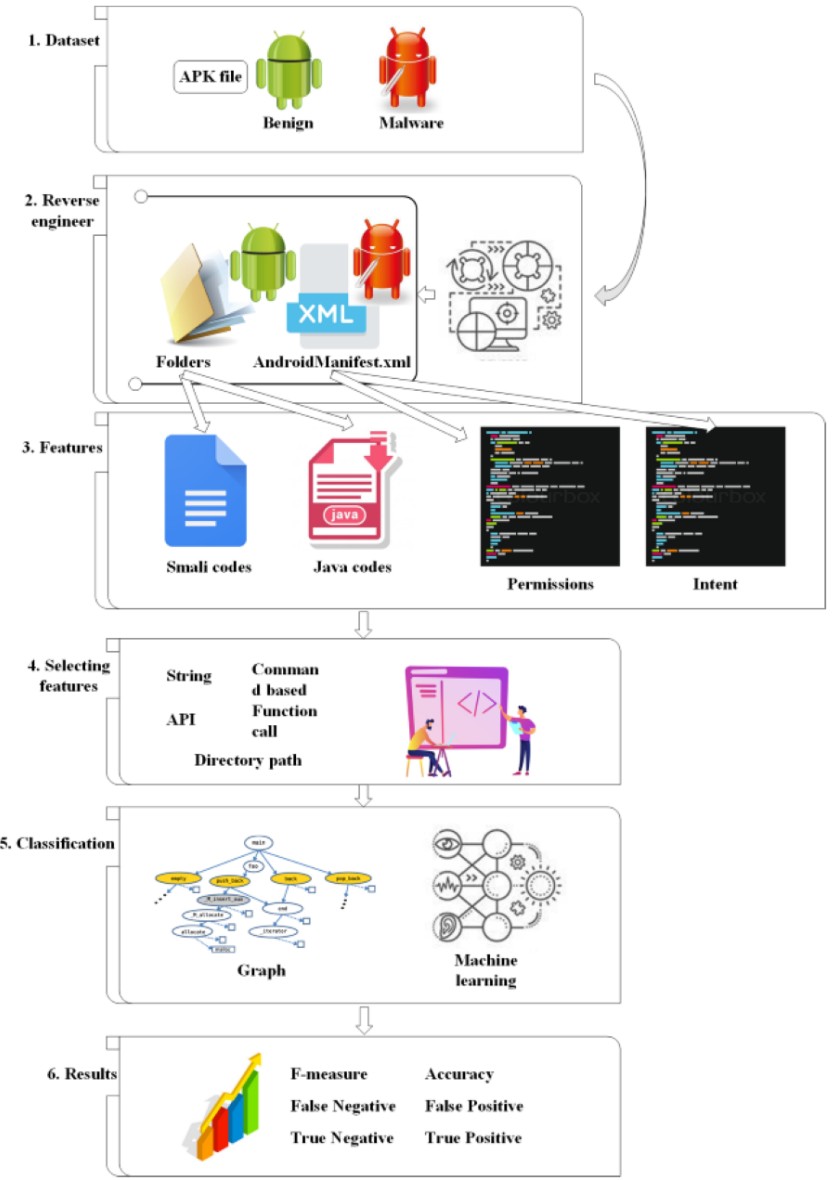

**Figure 2 Malware detection using static analysis.** The static analysis operation, which consisted of several steps. The steps included dataset collections, reverse engineer, features identification, and classification.

each with the (.apk) filename extension. This was followed by the reverse engineering performed on these applications to retrieve the code by extracting a few folders from one .apk file, which consisted of nested files with codes (Java or smali). Furthermore, one .apk would comprise approximately a thousand lines of codes. Therefore, with a total of 1,000 applications in one dataset, the security practitioners were required to scrutinise millions of lines of code. With the completion of the reverse engineering, an analysis would be conducted, which involved features. Features consist of a series of application

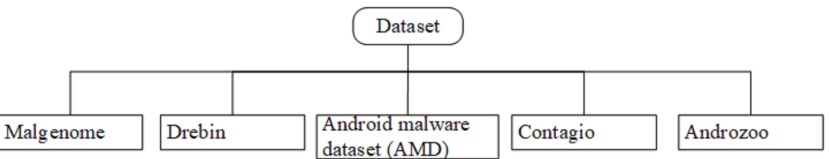

**Figure 3 Dataset of Android samples.** This dataset collected from five main dataset group samples. The datasets were used by the previous researchers for their works too.

characteristics for the detection of malware, while classification is an approach used to differentiate between malware and benign (normal) application. The following section thoroughly discusses the static analysis, which specifically begins with a focus on the dataset.

### Dataset

Figure 3 illustrates the Android malware dataset from different places. Notably, the majority of the datasets were obtained from universities. The datasets were in the form of an Android application package, which was followed by an .apk filename extension. Malgenome (*Anonymous, 0000d*) is the name of Android malware dataset, which was made to be publicly available with permission from their administrator. These malware samples, which were collected by North Carolina State University (NCSU) from August 2010 to October 2011, covered multiple families of malware consisting of botnet and root exploit. The characterization of the malware families was based on the method of the installation, the way the malware carried the malicious payloads, and its method of activation.

Androzoo (*Allix et al., 2018*; *du Luxembourg, 2016*) is another dataset consisting of approximately more than three million of Android applications (.apk). This dataset originates from the University of Luxembourg to contribute to the community for research purposes and further explore the notable development in the detection of malware, which damages the Android. Drebin (*Technische Universität Braunschweig, 2016*) dataset also presents Android malware publicly with strict requirements. A university from Germany (University in Braunschweig, Germany) collected 5,560 samples with 179 families. The time range provided for the malware was from August 2010 to October 2012. The university project, which was known as MobileSandbox, was an initiative for the acquirement of samples for academia and industry.

Android malware dataset (AMD) is a public Android malware dataset from the University of South Florida, which consists of 24,650 samples with 71 categorised families. To obtain this dataset, the user is required to acquire permission from the university and provide authentic information with evidence. The academia and the industry are allowed to use these samples for research purposes.

Contagio (*MilaParkour, 2019*) dataset presents the malware, which focuses on mobile malware, with a condition that the user should omit one sample to obtain another sample. It provides a dropbox for the user to share their mobile malware samples. According to their blogspot (*MilaParkour, 2019*), the name of the administrator of this dataset is Mila Parkour, who is reachable only through emails. Based on Table 5, which presents the

**Table 5  Previous article with the use of different datasets.** Presents the research articles and the respective datasets, and it could be seen that the dataset providers receive significant attention from other universities and the industry.

| Dataset | References of the articles with the use of the respective datasets |
|---|---|
| Malgenome *Anonymous (0000d)* | *Yerima, Sezer & McWilliams (2014)*, *Firdaus et al. (2017)*, *Firdaus & Anuar (2015)*, *Firdaus et al (2018)* |
| Drebin *Anonymous (0000c)* | *Firdaus et al. (2017)*; *Firdaus et al. (2018)*; *Firdaus et al. (2018)* |
| Android malware dataset (AMD) | *Badhani & Muttoo (2019)* |
| Contagio *MilaParkour (2019)* | *Feldman, Stadther & Wang (2014)*; *Islamic & Minna (2015)* |
| Androzoo *Université du Luxembourg (2018)* | *Razak et al. (2019)*; *Firdaus et al. (2017)*; *Razak et al. (2018)*; *Firdaus et al. (2017)* |

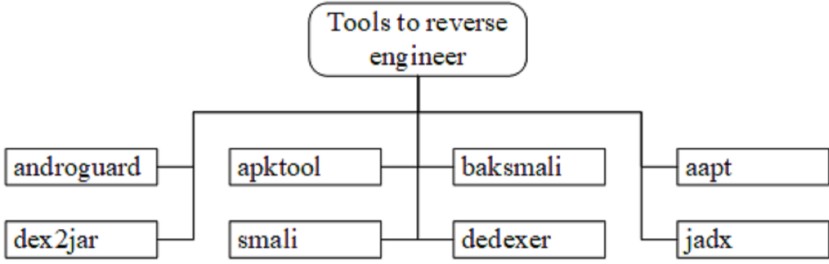

**Figure 4  Reverse engineer tools for static analysis.** This is the example of reverse engineer tools that have been used by the previous researchers to extract the code for malware.

research articles and the respective datasets, it could be seen that the dataset providers receive significant attention from other universities and the industry. It is hoped that this action would enhance the security of the Android device and its users from time to time.

### Reverse engineer

Static analysis is an activity to investigate the code of an application without executing it. In order to investigate, security practitioners implement the reverse engineering method. This method reversed from the executable file to its source code (*Dhaya & Poongodi, 2015*). This reverse engineering process loads the executable into a disassembler to discover what the program does. Figure 4 illustrates the tools used to perform a reverse engineering method, which was also adopted by security practitioners to identify Android malware. Table 6 illustrates the tools adopted in the respective articles.

### Features

Once the researchers reverse engineer the executable file using specific tools, they need to select features from the source code. Feature selection is important in order to increase the accuracy of the detection system (*Feizollah et al., 2015*; *Chanda & Biswas, 2019*; *Klaib, Sara & Hasan, 2020*) Figure 5 presents the taxonomy of multiple static features. The next sections are the details for each type of static feature.

*Advertisement libraries.*  Provided that most Android applications are available for free download, Android developers need to include advertisement libraries (ad libraries) in

**Table 6  Previous articles and the respective reverse engineer tools.** The tools for reverse engineer adopted in the respective articles are listed in this table.

| Tools | References to the articles and the respective tools |
|---|---|
| apktool | *Wiśniewski (2010)*; *Deshotels, Notani & Lakhotia (2014b)*; *Wu et al. (2012)*; *Faruki et al. (2013)*; *Luoshi et al. (2013)* |
| aapt | *Android (2013)*; *Sanz et al. (2013)* |
| androguard | *Desnos (2012a)*; *Suarez-Tangil et al. (2014)*; *Sahs & Khan (2012)*; *Aafer, Du & Yin (2013)*; *Junaid, Liu & Kung (2016a)* |
| baksmali | *Anonymous (2019b)*; *Apvrille & Strazzere (2012)*; *Huang, Tsai & Hsu (2012)*; *Grace et al. (2012b)*; *Zhou et al. (2013)*; *Zhou et al. (2012)*; *Grace et al. (2012a)* |
| dex2jar | *Anonymous (2019d)*; *Huang et al. (2014a)*; *Sheen, Anitha & Natarajan (2015)*; *Lee & Jin (2013)*; *Luoshi et al. (2013)* |
| jadx | *Skylot (2015)*; *Firdaus & Anuar (2015)* |
| dedexer | *Anonymous (0000b)* |
| smali | *Kang et al. (2013)*; *Anonymous (2019b)* |

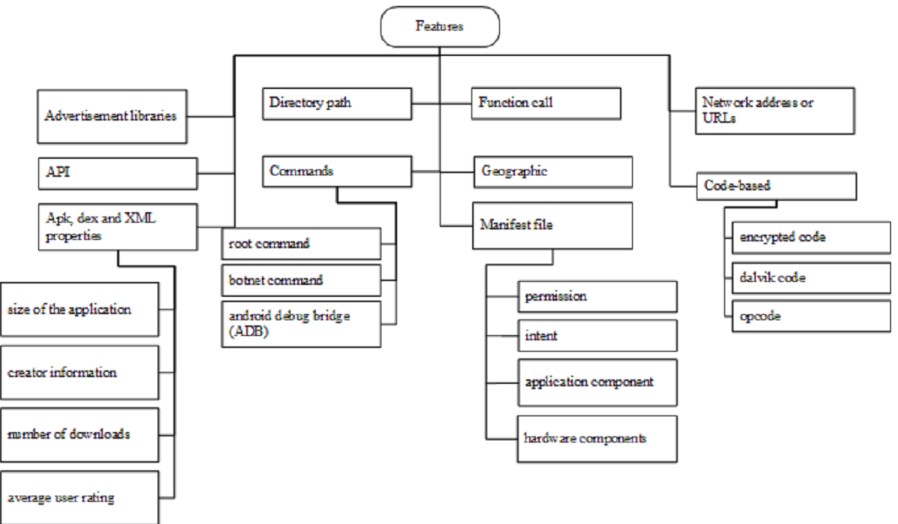

**Figure 5  Taxonomy of multiple static features.** Each static feature was figure out from the various experiments done using the specific tools and methods.

the free application for financial purposes. During the run-time of the application, the ad libraries would transfer the data regarding users' activities. The developer would then receive an incentive based on certain metrics of the information. Adrisk (*Grace et al., 2012a*) scrutinised and measured the risk of the codes of the ad libraries to detect applications, which may harm the users.

*Application programming interface.*  Application program interface (API) is a set of code ready for certain functionalities. Android application developers use this API for their application. Usually, there is documentation ready to use the API. Nevertheless, the attacker uses certain API for their malware application. Accordingly, to detect malware,

security practitioners inspect API features that regularly used by the attackers. The articles that involves API features are Droidlegacy (*Deshotels, Notani & Lakhotia, 2014b*), Droidapimiiner (*Aafer, Du & Yin, 2013*), Warning system (*Lee & Jin, 2013*; *Chang & Hwang, 2007*; *Wu et al., 2012*; *Bartel et al., 2012*; *Grace et al., 2012b*; *Wu et al., 2012*; *Shuying Liang et al., 2013*; *Zhou et al., 2013*; *Feng et al., 2014*; *Huang et al., 2014*; *Steven Arzt et al., 2014*; *Arp et al., 2014*; *Luoshi et al., 2013*; *Yerima, Sezer & Muttik, 2014*; *Seo et al., 2014*; *Sheen, Anitha & Natarajan, 2015*; *Peiravian & Zhu, 2013*).

*Apk, dex and xml properties.* Several security practitioners adopted the features, which consist of .apk file properties. The authors of the malwares (*Kang et al., 2015*) and (*Zhou et al., 2012*) are examined in two experiments due to the significant number of Android malwares written by a similar person. Therefore, the features of the malwares include serial numbers of the author, author's information, name, contact and organization information, developer certification, author's ID, and public key fingerprints of the author. Other features highlighted in this section are the application name, category, package, description, rating values, rating counts, size, number of zip entries, and common folders (*Samra, Kangbin & Ghanem, 2013*; *Shabtai, Fledel & Elovici, 2010*).

*Directory path.* Directory path allows access for a specific folder in the operating system (OS). It was found by security practitioners that the attacker incorporated a directory path for a sensitive folder in their malware. Meanwhile, several paths related to Android kernel directory were identified by another study (*Firdaus & Anuar, 2015*), such as 'data/local/tmp/rootshell', '/proc', and '/system/bin/su'.

*Commands.* Two types of commands are available, namely (1) root command and (2) botnet command. Specifically, several root commands were identified by (*Firdaus & Anuar, 2015*) in the Unix machine, such as 'cp', 'cat', 'kill', and 'mount'. Normally, these commands were used by the administrators to execute higher privileged actions in the Unix machine. Provided that Android architecture was based on the Unix kernel, the attackers included root commands in their malware to control the victim's Android devices. Therefore, the identification of root commands is crucial in investigating malwares.

The second type of command is a botnet command. Meanwhile, one type of malware, which is known as a mobile botnet, includes botnet commands in their malware codes, such as 'note', 'push', 'soft', 'window', 'xbox', and 'mark'. The attacker used these commands to communicate with the command and control (C&C) server, while droidanalyzer (*Seo et al., 2014*) combines API, root command, and botnet command into a set of features to detect root exploit and mobile botnet.

Other than ad libraries, certain researchers inspect the Android Debug Bridge (adb) code. ADB (*Android Developers, 2017*) is a tool, which provides a command-line access facility for users or developers to communicate with Android mobile devices. This facility allows the installation of unwanted applications and execution of various Unix by the attacker in the victim's device. Therefore, RODS (*Firdaus et al., 2018*) is a root exploit detection system for the detection of a root exploit malware with ADB features.

*Function call.* In programming, a function call is a declaration, which consists of a name and is followed by an argument in parenthesis. The list of the argument may include any numbers of the name, which are either separated by commas or left empty. Another study by *Aubrey-Derrick Schmidt et al. (2009a)* involved the extraction of a function call through readelf, which was then used for the features in machine learning prediction. Meanwhile, *Gascon et al. (2013)* extracted the function calls in a graph to identify the nodes from the start to the end of the process.

*Geographic location.* Geographic location is a feature, which identifies the origin of the application. The geographic detector was identified as one of the features in research by (*Steven Arzt et al., 2014*). Provided that 35% of the mobile malware families appeared to originate from China with 40% of the facilities originating from Russia, Ukraine, Belorus, Latvia, and Lithuania countries, it was crucial to consider geographic location as one of the features for the detection of Android malware. For this reason, researchers increased the risk signal for the applications originating from the aforementioned countries.

*Manifest file.* Android application is built on the top of the application framework which provides an interface for the user. The program is based on the Android application package file in the (.apk) format, which is also used to install an application in android-based mobile devices. It consists of meta-inf, resource, assets and library directory, classes.dex, resources.arsc, and androidmanifest.xml file. One of the files, androidmanifest.xml (manifest file), is an essential file with contents of various features, such as permission, intent, hardware component, and components of the application (activities, services, broadcast receivers, and content providers) (*Android, 2015*).

(a) Permission

Permission is a unique security mechanism for Android devices. To enable the permission, the user needs to allow the application during the installation period. However, many users accidentally enable certain permissions, which leads to access to sensitive security-relevant resources. Therefore, permission features were examined in many studies. Based on the application of permission in several studies to measure the risk of the application, permission was further identified as malicious (*Razak et al., 2018*; *Razak et al., 2019*). Some other studies, such as (*Hao Peng et al., 2012*; *Samra, Kangbin & Ghanem, 2013*; *Walenstein, Deshotels & Lakhotia, 2012*; *Huang, Tsai & Hsu, 2012*; *Sahs & Khan, 2012*; *Sanz et al., 2013*; *Talha, Alper & Aydin, 2015*; *Aung & Zaw, 2013*), used the permission features as the inputs for machine learning prediction.

(b) Intent

The intent is coded in the manifest file and allows a component of the application to request certain functionality from another component from other application. For example, application A can use the component of application B for the management of photos in the device despite the exclusion of the component from application A. Provided that this feature enables malicious activities among the attackers, several experiments used intent (declared in the manifest file) as one of the features for the detection of malware, such as (*Feizollah et al., 2017*; *Fazeen & Dantu, 2014*).

(c) Application component

The manifest file declared application component, which consists of four types, namely (1) activities, (2) services, (3) broadcast receivers, and (4) content providers. Specifically, activity is represented as the user interface or interactive screen to the users, while service refers to an operation occurring in the backgrounds, which perform the long-service process. This is followed by broadcast receivers, which respond to system-wide broadcast announcements. On the other hand, content providers manage a structured set of application data. Overall, these four components follow a life cycle model during execution. Dexteroid (*Junaid, Liu & Kung, 2016*) proposed a framework, which systematically guides the event sequences through the reverse engineering/reconstruction of the life cycle models and the extraction of callback sequences from event sequences to detect malicious behaviours.

(d) Hardware component

The manifest file also incorporated hardware components in the Android application. To illustrate, the developer requested access to the camera of an Android device by declaring it in the manifest file to enable the use of the camera for the application. However, the attacker declared unrelated hardware components in their game application, such as camera and data. As a result, the security researchers were prompted to use hardware component as the features in their experiment (*Arp et al., 2014*) to detect malware (*Allahham & Rahman, 2018*).

*Network address.* Access to the Internet is essential for attackers to retrieve private information of the victim, change the settings, or execute malicious commands. This process requires the incorporation of the Uniform Resource Locator (URL) or network address in the malware code. The examples of sensitive URLs include the Android Market on Google Play, Gmail, Google calendar, Google documents, and XML schemas. These features were used in *Luoshi et al. (2013)* and *Apvrille & Strazzere (2012)*, *Mohd Azwan Hamza & Ab Aziz (2019)* for malware detection.

*Code-based.* Code-based or code structure comprises a line or set of programming language codes in an application. Two studies applied code structures (code chunk grammar) as the features for malware detection, which is focused on the internal structure of the code units (*Suarez-Tangil et al., 2014*; *Atici, Sagiroglu & Dogru, 2016*). This feature enables the analysis and differentiation between malware and benign applications. Another study by *Firdaus & Anuar (2015)* identified several code-based strings, namely '.exec', 'forked', 'setptywindowsize', and 'createsubprocess'. In comparison with the normal application, it was found that the attacker frequently used these code-based features in the development of malware. Therefore, these features were also used in this study to detect malware.

Opcode (operation code) is another code-based feature. It is a part of the instruction to inform the CPU regarding the tasks to be fulfilled. Assembly language used this opcode to execute the instruction. Also referred to as bytecode, the examples of an opcode for Android included OP_ADD_DOUBLE, OP_ADD_FLOAT, OP_ADD_INT_2ADDR, and OP_SUB_LONG (*Developer, 2020*). Specifically, this feature was adopted in the studies

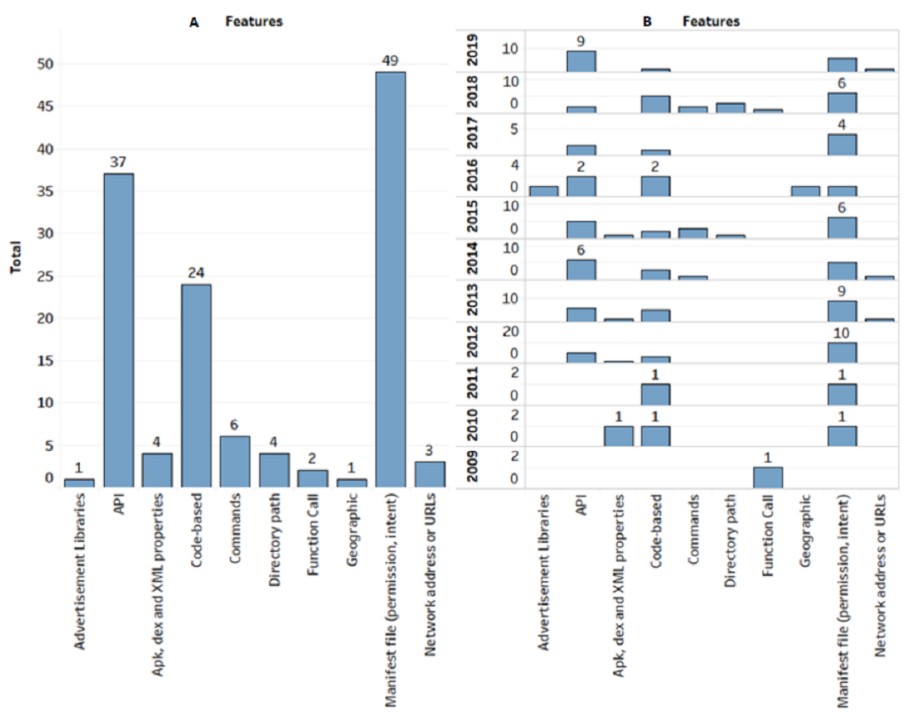

**Figure 6 Categories of features in total and years.** (A) The graph shows the features identified using static analysis, depicts that researchers prefer to investigate permission and API features compare to others. (B) Total number of features based on years.

by *Zheng, Sun & Lui (2013)*, *Medvet & Mercaldo (2016)*, *Faruki et al. (2013)* and *Zhao et al. (2019)* to detect Android malware in the static analysis. Further examples of the features in this section are method (*Kim et al., 2018*), opcode (*Zhao et al., 2019*), byte stream @ byte block (*Faruki et al., 2013*), Dalvik code (*Gascon et al., 2013*), and code involving encryption (*Gu et al., 2018*). The selection of the features by security practitioners is followed by classification. This process was performed to receive the features as input and differentiate between either the application malware or benign (normal).

Figure 6 depicts that researchers prefer to investigate permission and API features compare to others. However, the trend in permission features is decline from 2013 until 2018. However, API features takes place in previous experiments as it increased from six (2014) to 9 (2019). This indicates that the API trend would increase in following year in static detection.

### Classification

In the classification process for static analysis, many security analysts used two types of methods; (1) Machine learning (ML) and (2) Graph. The following section presents the ML studies with static features.

*Machine learning (ML).* Machine learning is a scientific discipline, which is capable to predict future decisions based on the experience it has gained through past inputs (learning

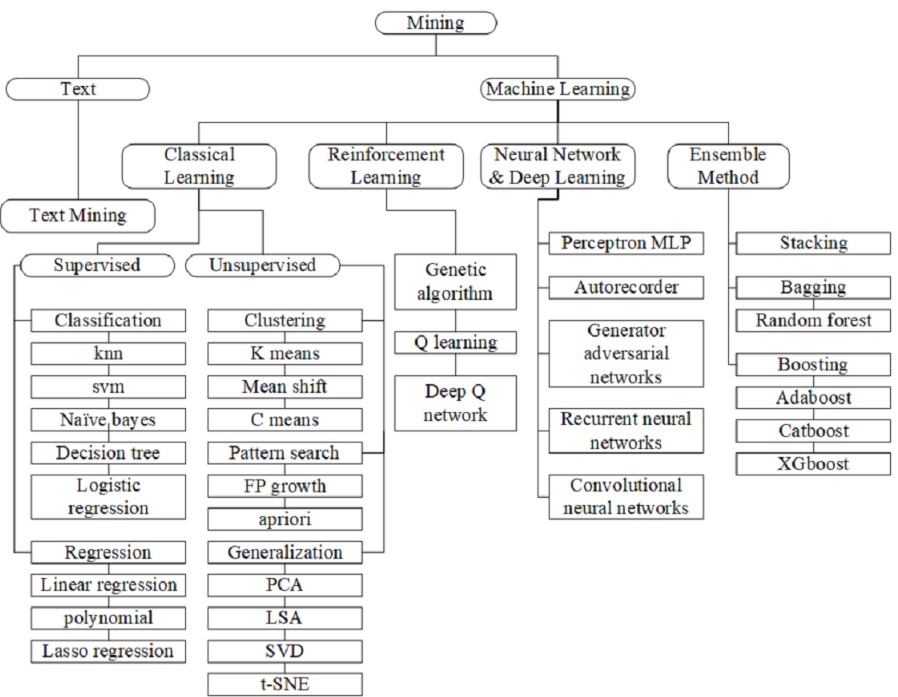

**Figure 7 Mining in static analysis.** A few types of machine learning available, with each type, has its own classifier as listed in figure.

set), followed by a prediction of the outputs. Basing on a given dataset, the learning set makes intelligent decisions according to certain algorithms. One of the machine learning types is supervised based on the data for the training stage to create a function. Furthermore, each part of the training data contains input (features or characteristics) and output (class label-malware and benign). This is followed by the training stage, which calculates the approximate distance between the input and output examples to create a model. This training stage could classify unknown applications, such as malware or benign application. Four types of ML are present, such as (1) classical learning; (2) reinforcement learning, (3) neural network and deep learning, and (4) ensemble method. Figure 7 illustrates the ML taxonomy, which starts with classical learning.

(a) Supervised Learning

Supervised learning (SL) is a process of learning from previous instances to predict future classes. Therefore, the prediction of the class label involves the construction of a concise model from previous experience. The machine learning classifier is then used to test the unknown class (*Kotsiantis, 2007*). To detect Android malware with static features, the SL method is widely used by security practitioners. Accordingly, the previous articles adopting this method are illustrated in Table 7.

(b) Unsupervised Learning

Unsupervised learning is another type of learning involved in machine learning. It is a clustering technique where the data is unlabeled and has also been used in computer security areas, including malware detection and forensic (*Beverly, Garfinkel & Cardwell,*

**Table 7  Machine learning and its classifier used in studies.** Machine learning types supervised and unsupervised with the classifier and the respective articles. To detect Android malware with static features, the supervised learning method is widely used by security practitioners.

| Machine learning type | Classifier | Reference |
|---|---|---|
| Supervised | K-nearest neighbor | *Anonymous (2019b)*; *Aafer, Du & Yin (2013)* |
| | Support vector machine (svm) | *Aafer, Du & Yin (2013)*; *Sahs & Khan (2012)*; *Huang, Tsai & Hsu (2012)*; *Arp et al. (2014)*; *Anonymous (2019b)*; *Anonymous (2019d)*; *Arp et al. (2014)* |
| | CART | *Aung & Zaw (2013)* |
| | Adaboost | *Huang, Tsai & Hsu (2012)*; *Sheen, Anitha & Natarajan (2015)* |
| | Bayes | *Yerima, Sezer & McWilliams (2014)*; *Lee, Lee & Lee (2015)*; *Hao Peng et al. (2012)*; *Sanz et al. (2013)*; *Shabtai, Fledel & Elovici (2010)* |
| | Logistic Regression | *Talha, Alper & Aydin (2015)* |
| | Prism (PART) | *Aubrey-Derrick Schmidt et al. (2009a)*; *Yerima, Sezer & Muttik (2014)*; *Shabtai, Fledel & Elovici (2010)* |
| | Voting feature interval (vfi) | *Shabtai, Fledel & Elovici (2010)* |
| | Random forest | *Shabtai, Fledel & Elovici (2010)*; *Sanz et al. (2013)*; *Aafer, Du & Yin (2013)*; *Huang, Tsai & Hsu (2012)* |
| | Sequential minimal optimisation (smo) | *Sanz et al. (2013)* |
| | Instance-based learning with parameter k (ibk) | *Sanz et al. (2013)* |
| | Simple logistic | *Sanz et al. (2013)* |
| | Multilayer perceptron | *Firdaus & Anuar (2015)* |
| Unsupervised | K-means | *Fan et al. (2019)*; *Wu et al. (2012)*; *Samra, Kangbin & Ghanem (2013)*; *Aung & Zaw (2013)* |
| | Normalised Compression distance (NCD) | *Lu et al. (2012)*; *Crussell, Gibler & Chen (2012)* |

*2011*). Clustering refers to the division of a large dataset into smaller data sets with several similarities. It classifies a given object set through a certain number of clusters (assume *k* clusters) to determine the *k* centroids assigned for each cluster. In this case, this algorithm selects the centroid by random from the applications set, extracts each application from a given dataset, and assigns it to the nearest centroid. Table 7 tabulates the previous articles, which adopted this method.

(c) Reinforcement learning

A reinforcement learning model consists of an agent (a set of actions A) and an environment (the state space S) (*Anderson et al., 2018*). Deep reinforcement learning was introduced by reinforcement agents as a framework to play Atari games, which often exceed human performance (*Volodymyr Mnih et al., 2013*; *Volodymyr et al., 2015*). The advances in deep learning may extract high-level features from raw sensory data, leading to breakthroughs in computer vision and speech recognition. In the case of deep learning, the agent would be required to learn a value function in an end-to-end way, which takes raw pixels as input and predicts the output rewards for each action.

The learned value function is called deep Q learning, in which Q function is learned and refined from over hundreds of games (*Anderson, Filar & Roth, 2017*). The Q-learning algorithm was trained in network (*Volodymyr Mnih et al., 2013*) with stochastic gradient descent to update the weights. Replay mechanism was used from random samples previous transitions to lead smooth training distribution over past behaviors to overcome the correlated data and non-stationary distributions problems. *Anderson et al. (2018)* propose a framework based on reinforcement learning (RL) for attacking static portable executable (PE) anti-malware engines. Meanwhile, a DQN-based mobile proposed by *Wan et al. (2018)* to enhance the malware detection performance. The results shown from simulation can increase the malware detection accuracy and reduce the detection delay as compared to a Q-learning based malware detection scheme.

(d) Neural Network and Deep Learning

The evolution of Neural Network (NN) has been associated with various challenges since the mid-20th century. McCulloch and Pitts obtained the first inspiration of NN in 1943 from biological neurons, which was followed by proposing a computational model for the development of hypothetical nets. Although this proposal was simulated by Nathaniel Rochester at IBM research laboratory, this attempt was unsuccessful at the end. Developed by Frank Rosenblatt at Cornell Aeronautical Laboratory, the perceptron became the first learning machine (*Mahdavifar & Ghorbani, 2019*).

Despite all the upgrades on NNs, Deep learning (DL) was developed in 2006 and has been used in almost every application. As a new variation of the classical Multilayer Perceptron (MLP), the DL aims to produce high-level and flexible features from the raw pixel data to assist in generalising the classification. Furthermore, DL also operates with complex applications containing millions of data, which require a large number of neurons and hidden layers. A few DL frameworks have been developed in the recent years, such as TensorFlow (*Abadi, Agarwal & Barham, 2016*), Caffe (*Yangqing Jia et al., 2014*), and Theano (*Al-Rfou et al., 2016*) to ensure an efficient implementation of Deep Network (DN) architectures and omit the unnecessary coding scratching (*Mahdavifar & Ghorbani, 2019*). Additionally, the DL method extracts the features based on the layer's level, which could either be high or low.

Figure 8 depicts the differences between ML and DL. It shows that ML requires the security practitioners to extract the features manually and select the ML classifier, which is suitable for the selected features. However, DL involves automatic feature extraction part and malware classification. It trains the model end-to-end with the Android application package (.apk) file and their categories, each labelled as malware or benign. The DL gains and creates a prediction model through the automatic selection of the feature.

As one of the major models in deep learning, a convolutional neural network (CNN) has been widely used for image recognition (*Li et al., 2018*). It could be seen in the past few years that many studies have implemented Deep Neural Networks (DNN) to classify malware (*Pascanu et al., 2015*; *Saxe & Berlin, 2016*; *Zhao et al., 2019*). Additionally, although the recurrent neural networks have been explored since the 1980s, they have become uncommercial due to several issues (*Pascanu et al., 2015*). Several machine learning methods have addressed network or malware attacks on personal computers or mobile

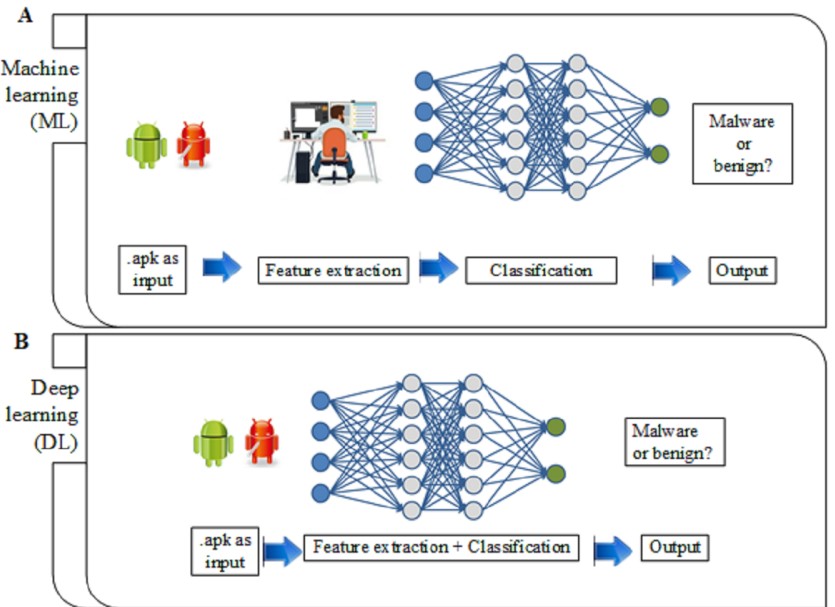

**Figure 8** **Differences between ML and DL.** (A) ML extract the features manually and select the ML classifier. (B) DL involves automatic feature extraction and malware classification.

devices. Simultaneously, several techniques were proposed by researchers who applied DL algorithms to detect or categorize malware using static, dynamic, or hybrid approaches, detection of network intrusions and phishing/spam attacks, and inspection of website defacements (*Venkatraman, Alazab & Vinayakumar, 2019*).

(e) Ensemble method

Another technique in machine learning and pattern recognition is ensemble learning. The increase in the implementation of ensemble learning methods could be seen in the computational biology field due to the unique advantages in managing small sample size, high dimension, and complex data structures (*Yang et al., 2010*). The function of ensemble learning is to build a prediction model by combining the strengths of the collection of simpler base models (*Hastie, Tibshirani & Friedman, 2009*). A few approaches are applied in ensemble methods, such as bagging, boosting, and random forest. This method is also a simple device, which is popular especially in the predictive performance of a base procedure.

The bagging procedure appears to be a variance reduction scheme for some base procedure, while the boosting methods mainly reduce the bias of the base procedure. Therefore, the significant difference between bagging and boosting ensemble methods is indicated. Compared to bagging and boosting, the random forest approach is a highly distinguished ensemble method. The first proposal of the random forest was made by *Amit & Geman (1997)*. While the performance of random forests is on the same level as boosting, it could exhibit better performance in the perspective of prediction.

Table 8 shows previous works done using different types of machine learnings as mentioned before. From the table, we can summarize classical learning is still valid to be

**Table 8 Static analysis works using various types of machine learning.** We can summarize classical learning is still valid to be used in experiment execution but there are a lot of works are using deep learning and graph method in the current trend.

| Year | Ref | Machine Learning used (✓) | | | | | Metrics |
|---|---|---|---|---|---|---|---|
| | | Classical learning | Reinforcement learning | Neural network & deep learning | Ensemble method | Others | |
| 2019 | *Yildiz & Doğru (2019)* | ✓ | | | | | Support Vector Machine (SVM) |
| 2019 | *Singh, Jaidhar & Kumara (2019)* | ✓ | | | | | Linear Support Vector Machine (L-SVM) |
| 2019 | *Lei et al. (2019)* | | | ✓ | | | k-means |
| 2018 | *Firdaus et al. (2018)* | | | | ✓ | | Logitboost |
| 2018 | *SL et al. (2015)* | | ✓ | | | | Hotbooting-Q |
| 2018 | *Yangqing Jia et al. (2014)* | | | ✓ | | | Stochastic gradient descent (SGD) |
| 2018 | *Firdaus et al. (2017)* | | | ✓ | | | Multilayer Perceptron (MLP), Voted Perceptron (VP) and Radial Basis Function Network (RBFN). |
| 2018 | *Firdaus et al. (2018)* | | | ✓ | | | Naïve Bayes (NB), functional trees (FT), J48, random forest (RF), and multilayer perceptron (MLP) |
| 2018 | *Li et al. (2018)* | | | ✓ | | | Bayesian calibration |
| 2017 | *Anderson, Filar & Roth (2017)* | | ✓ | | | | Direct gradient-based, White-box, Binary black-box |
| 2017 | *Volodymyr Mnih et al. (2013)* | | ✓ | | | | Black-box |
| 2016 | *Junaid, Liu & Kung (2016)* | | | | | ✓ | Control flow graph |
| 2016 | *Wan et al. (2018)* | | | ✓ | | | FCNN, DistBelief |
| 2016 | *Abadi, Agarwal & Barham (2016)* | | | ✓ | | | FCNN |

*(continued on next page)*

**Table 8** (*continued*)

| Year | Ref | Machine Learning used (✓) | | | | | Metrics |
|------|-----|---------------------------|---|---|---|---|---------|
| | | Classical learning | Reinforcement learning | Neural network & deep learning | Ensemble method | Others | |
| 2015 | *Pascanu et al. (2015)* | | | ✓ | | | Sequential Minimal Optimization (SMO), Support Vector Machine (SVM) |
| 2015 | *Kang et al. (2015)* | ✓ | | | | | Naïve Bayes |
| 2015 | *Talha, Alper & Aydin (2015)* | ✓ | | | | | Support Vector Machine (SVM) |
| 2015 | *Anderson et al. (2018)* | | ✓ | | | | Deep Q-network |
| 2015 | *Al-Rfou et al. (2016)* | | | ✓ | | | Echo state networks (ESN) |
| 2014 | *Deshotels, Notani & Lakhotia (2014a)* | | | | | ✓ | Class Dependence Graph (CDG) |
| 2014 | *Feng et al. (2014)* | | | | | ✓ | Inter-Component Call Graph |
| 2014 | *Huang et al. (2014)* | | | | | ✓ | Control flow graph (CFG) and call graph (CG) |
| 2014 | *Steven Arzt et al. (2014)* | | | | | ✓ | Inter-procedural control-flow graph (ICFG) |
| 2014 | *Mahdavifar & Ghorbani (2019)* | | | ✓ | | | FCNN, MNIST digit |
| 2013 | *Shuying Liang et al. (2013)* | | | | | ✓ | Entry-Point Saturation (EPS) |
| 2013 | *Zhou et al. (2013)* | | | | | ✓ | Vantage Point Tree (VPT) |
| 2013 | *Beverly, Garfinkel & Cardwell (2011)* | | ✓ | | | | Q-learning |
| 2012 | *Amit & Geman (1997)* | | | | | ✓ | Program dependence graphs (PDGs) |
| 2010 | *Saxe et al. (2015)* | | | | ✓ | | Jaccard set-based index |

used in experiment execution but there are a lot of works are using deep learning and graph method. The current trends show the demand using the deep learning technique to defend against an increasing number of sophisticated malware attacks where deep learning based have become a vital component of our economic and national security. Many recent studies on Android malware detection have leveraged graph analysis as mentioned in the next section.

*Graph.* The use of a graph is another method in machine learning and pattern recognition, which is performed by investigating the data and control-flow analysis. It is also capable of identifying unknown malware through the examination on the flow of the code. This method is preferred by security analysts due to the uniform flow despite the changes made by the malware authors on the API calls to avoid intrusion detection systems. The types of analysis in graph method include call graph, inter-component call graph (ICCG), control-flow graph (CFG), and dependence graph, while Table 9 lists the previous works of research on static malware detection using the graph method.

A call graph (specifically known as flow graph) is a graph representing the control and data flow of the application, which investigates the exchange of information through the procedures. A node in the graph represents a procedure or function, as seen from the x and y symbols, which indicate that procedure x calls for procedure y. Apposcopy (*Feng et al., 2014*) presents its new form of call graph known as inter-component call graph (ICCG) to match malware signature. As a directed graph where nodes are known as components in an application, it establishes ICCG from a call graph and the results of the pointer analysis. The objective of apposcopy is to measure the inter-component communication (ICC), calls, and flow relations.

Another graph called a control flow graph (CFG) is also applied by many security analysts to investigate the malware programme. Woodpecker (*Grace et al., 2012b*) created the CFG start from each entry point (activity, service, receiver, content provider), which is defined in the permission stated in the androidmanifest.xml file. Furthermore, the public interface or services from an execution path is discovered through the flow graph. However, it would be considered by Woodpecker as a capability leak if it is not guarded by the permission requirement nor prevented from being invoked by another unrelated application. The same graph was applied in subsequent works of research, namely Flowdroid (*Steven Arzt et al., 2014*), Dendroid (*Suarez-Tangil et al., 2014*; *Sahs & Khan, 2012*), Asdroid (*Huang et al., 2014a*), Anadroid (*Shuying Liang et al., 2013*), Adrisk (*Grace et al., 2012a*), and Dexteroid (*Junaid, Liu & Kung, 2016a*).

Another graph is the dependency graph, which illustrates the dependencies of several objects on each other. An example could be seen in the dead code elimination case process, in which the graph identifies the dependencies between operation and variables. With the dependency of non-operation on certain variables, these variables would be considered dead and should be deleted. The studies, which adopted this type of graph are CHEX (*Lu et al., 2012*), Dnadroid (*Crussell, Gibler & Chen, 2012*), Droidlegacy (*Deshotels, Notani & Lakhotia, 2014b*; *Zhou et al., 2013*).

**Table 9  Previous static analysis research, which used the graph method.** The types of analysis in graph method include call graph, inter-component call graph (ICCG), control-flow graph (CFG), and dependence graph.

| Type of graph | Reference |
|---|---|
| Call graph | Copes (*Bartel et al., 2012*), Leakminer (*Yang & Yang, 2012*), Riskranker (*Grace et al., 2011*), A3 (*Luoshi et al., 2013*) and (*Gascon et al., 2013*) |
| Inter-component call graph (ICCG) | *Feng et al. (2014)* |
| Control flow graph (CFG) | Woodpecker (*Grace et al., 2012b*), Flowdroid (*Steven Arzt et al., 2014*), Dendroid (*Suarez-Tangil et al., 2014*; *Sahs & Khan, 2012*), Asdroid (*Huang et al., 2014a*), Anadroid (*Shuying Liang et al., 2013*), Adrisk (*Grace et al., 2012a*), and Dexteroid (*Junaid, Liu & Kung, 2016a*) |
| Dependency graph | CHEX (*Lu et al., 2012*), Dnadroid (*Crussell, Gibler & Chen, 2012*), Droidlegacy (*Deshotels, Notani & Lakhotia, 2014b*) and (*Zhou et al., 2013*) |

*Others.* Besides machine learning and graph, several security practitioners adopted different methods, such as Normalized Compression Distance (NCD). Adopted in the studies by *Desnos (2012b)* and *Paturi et al. (2013)*, this method can measure the similarities between the malwares and represent them in the form of a distance matrix. Despite the evolution of many malwares from time to time, some of their behaviour patterns are similar to each other. The calculation of the similarities using NCD would identify the malwares, which share the same distance.

A study known as DelDroid (*Hammad, Bagheri & Malek, 2019*) implemented a method called as Multiple-Domain Matrix (MDM). This method refers to a complex system, which calculates multiple domains and is based on the Design-Structure Matrix (DSM) model. Furthermore, MDM is formed by the connection of DSM models with each other. The study initialised multiple domains in the MDM to represent the architecture of an Android system for privilege analysis. To illustrate, the incorporation of certain definitions in the MDM representation in the architecture enables DelDroid to identify the communication of the application, which may result in an unauthorised malware attack.

Another previous static experiment was conducted on the MD5 signature of the application to detect malware (*Seo et al., 2014*). In the first process, the study assigned the application as level C (the lowest level of suspicion), followed by calculation and cross-reference in the database of signatures. The application would be recorded if the result was positive. However, it would be identified as malware if the result of the suspicion was R. The system examined the files inside the application to find any matched MD5 signature.

Androsimilar (*Faruki et al., 2013*) practised a method known as a statistical similarity digest hashing scheme, which inspects the similarity on the byte stream based on robust statistical malicious static features. It is also a foot-printing method, which identifies the regions or areas of statistical similarity with known malware. Following that, it generates variable-length signatures to detect unknown malware (zero-day).

The following study is DroidMOSS (*Zhou et al., 2012*), which identifies between the repackaged (modified) and original application. This function is important due to the content of malicious activities in many Android repackaged applications. This study used a fuzzy hashing technique, which generated fingerprint based on this technique to localise and detect any previously applied modifications to the original application. It then

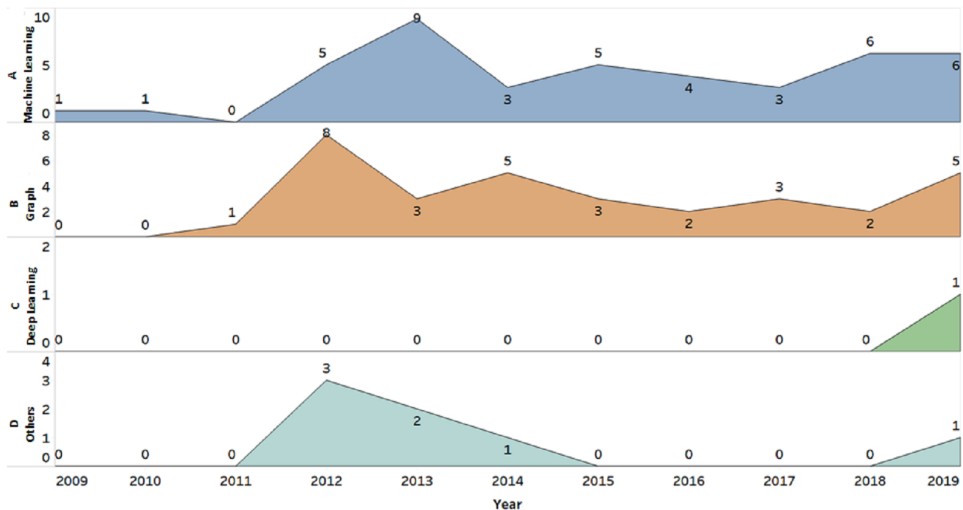

**Figure 9  Popular methods among security practitioners in static analysis.** Both ML and graph were the popular methods among security practitioners in static analysis. (A) ML was more preferred compared to graph. (B) The graph method was found to exceed the ML method in 2011, 2012, and 2014.

calculated the edited distance to measure the similarity between the applications. When the result of the similarity exceeds certain values, the application would be considered as a modified sample.

Under another static experiment, a study by *Apvrille & Strazzere (2012)* adopted a method known as a risk score weight, which was performed through the calculation of the risk score based on the selected features in the code. When the features were identified, the score increased according to certain risky patterns of properties. Particularly, the patterns were based on different likelihoods of the given situations between normal and malware samples. Lastly, the percentage of the likelihood was calculated. Figure 9 shows that both ML and graph were the popular methods among security practitioners in static analysis. The graph method was found to exceed the ML method in 2011, 2012, and 2014, although ML was more preferred compared to graph in other years. However, this situation reveals that graphs and ML are favourable options in the static experiment.

A study started to utilise DL (part of ML) in the static experiment in 2019, which also combined DL (Convolutional neural network—CNN) with Control flow graph (CFG). Notably, provided that API was the only feature utilised in this study, many future opportunities were available to combine different DL classifiers (Recurrent neural network—RNN, Generative* adversarial networks—GAN or Deep belief network*—DBN) with other features besides API and different types of graph. It is noteworthy that DL could also be combined with NCD and MDM.

## Open research issues

This section explains the issues involved in the static analysis, which were derived from previous research articles. Specifically, a discussion is made on the advantages and disadvantages of the open-source operating system, which rely on the availability of

the kernel application code. Another issue on static analysis is code obfuscation used by the malware developer to increase the difficulty to detect the malware and research done to counter obfuscation. Then, we review overall static analysis articles, how to detect unknown malware, the combination of static and dynamic, resource consumption, future work, and features.

### Open source advantages and disadvantages

Provided that Android malware is an open-source operating system, there is a continuous increase in its amount. To illustrate, one of the open-source availabilities is the kernel application code. Accordingly, Samsung officially provides its kernel operating system to the public (*Samsung, 2019*) for kernel enhancement or private purposes. Furthermore, any person may download the link according to the mobile device version. The code is also available in Lenovo (*Lenovo, 2021*), LG (*Official, 2019c*), Sony (*Sony, 2019e*), Htc (*HTC, 2019b*), Asus (*ASUS, 2019a*), Motorola (*Official, 2019d*), and other mobile providers. Consequently, this code availability may allow malware writers to identify and manage the vulnerabilities of the kernel's operating system.

Even though the availability of open-source contributes to its easy access, it is also available for security practitioners to research it. These practitioners may be the researchers of universities, the staff of the mobile providers, Android Google researchers, freelance programmers, and the Android community. They also invent various frameworks, algorithms, and suggestions to improve the security of the operating system. The version of the kernel is updated every year, while the mobile providers are informed to regarding the kernel updates. These studies, including static analysis, would increase the confidence of Android users worldwide.

### Obfuscation

Static analysis involves reverse engineering, such as decompile and disassemble, while malware developer utilises the obfuscation method to increase the difficulty of the decompiling process and lead to confusion in it. Obfuscation is a technique, which increases the difficulty in understanding the programmes due to the failure of the lead security analysts to distinguish between malware and benign application. Notably, it is a well-known obstacle to be examined by static analysis. Figure 10 illustrates the types of obfuscation, which include encryption, oligomorphic, polymorphism, and metamorphism (*Moser, Kruegel & Kirda, 2007*; *You & Yim, 2010*).

The encryption method was extensively practised by the malware writers. In this case, the important code or strings, which revealed the malware detector or security practitioner, should be identified. Accordingly, the code was encrypted and converted to the ciphertext. Furthermtore, various algorithms to encrypt the code are present, such as Caesar, Playfair, Data Encryption Standard (DES), Advanced Encryption Standard (AES), and Rivest-Shamir-Adelman (RSA). Therefore, for the security practitioner to understand the behaviour of the malware, the encrypted code should be decrypted using the correct decryptor (*Wei et al., 2017*).

Besides being a malware capable of mutating @ changing the decryptor, the oligomorphic is also able to generate multiple decryptors to hundreds of types (*You & Yim, 2010*).

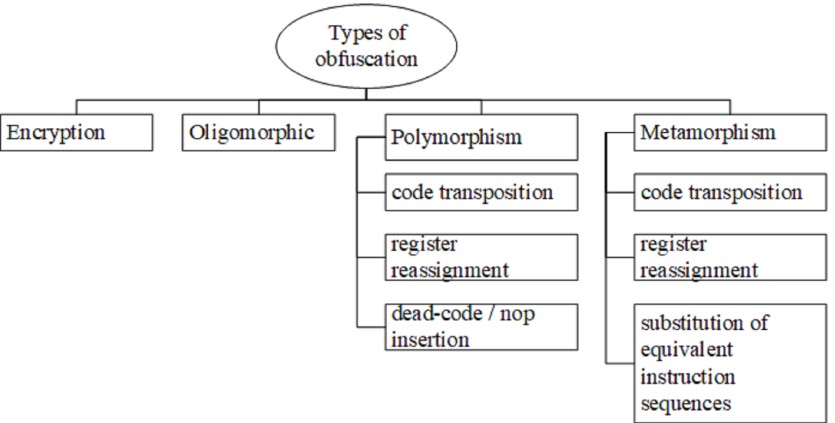

**Figure 10 Types of obfuscation.** Types of obfuscation identified include encryption, oligomorphic, polymorphism, and metamorphism.

Consequently, the security practitioner would need to change different decryptor multiple times until the code is returned to the normal string. Nevertheless, this type of obfuscation does not affect the size or shape of the code. Another type of obfuscation is polymorphic. It is a descriptor, which affects the size or shape of the code. Compared to oligomorphic, it is more advanced due to the incorporation of code transposition, register reassignment, dead code @ nop insertion, and armoring. Meanwhile, metamorphism is an approach beyond the oligomorphic and polymorphic types due to the absence of decryptor in its mechanism. Therefore, its constant body could be hidden from memory and increase the difficulty of the static investigation to detect malware.

The following information is the obfuscation methods that regularly used by polymorphism (polimorphic) and metamorphism (metamorphic) obfuscation (*You & Yim, 2010*).

(a) Code transportation

Code transposition is a method, which restructures the orders of the original code without causing any effects on its conduct. This process is performed with two methods. The first method is the random restructure of the original code by including jumps or unconditional branches. However, security practitioners can detect obfuscation by removing those jumps or unconditional branches. The second method is the production of new generations by selecting and restructuring independent instructions without any impact on others. However, the adoption of these methods is challenging for the malware writer, while the security practitioners are faced with a difficulty to detect this method of obfuscation.

(b) Register reassignment

Register reassignment is another method of obfuscation, which shifts the registers of the code from one generation to another. This method is performed without changing the behaviour of the code while keeping the programme of the code similar to its original state.

(c) Dead-code/nop insertion

Known as nop insertion, dead-code is a method, which adds several unnecessary instructions in the code and simultaneously keeps the behaviour of the code similar to its original state. Nevertheless, in certain situations, security practitioners able to detect this obfuscation by removing the aforementioned code.

(d) Substitution of equivalent instruction sequences

The original code is changed through the substitution of several instructions. To illustrate, the SUB instruction is changed to XOR, while PUSH is changed to MOV.

*Research to counter obfuscation.* To overcome obfuscation, many studies were conducted on different approaches. Study by *Crussell, Gibler & Chen (2012)* used program dependence graph (PDG) to prevent program transformations in obfuscation. Droidlegacy (*Deshotels, Notani & Lakhotia, 2014b*) use graph node to represents the java class in detecting light obfuscation. Droidanalytics (*Zheng, Sun & Lui, 2013*) and Drebin (*Arp et al., 2014*) extract the API calls while the codes running during execution time. In order to control the flow of obfuscation, Apposcopy use inter-component communication (ICC) to write the signature. Research by *Firdaus (2017)* uses jadx, one of reverse engineer tool to de-obfuscation the obfuscation codes. Summary of studies conducted to overcome obfuscation shown in Table 10.

*Advantage of obfuscation.* Despite the adoption of the obfuscation method by the malware writers or the attackers to evade detection, obfuscation also serves the following advantages based on other points of views:

(a) Reduction of the size of the application

Google (*Android, 2019b*) encourages developers to enable shrinking in their release to build an application to remove any unused codes and resources. Furthermore, provided that obfuscation would shorten the names of the classes and members in the code, the developer will be able to reduce the size of the application. Notably, the size of the application is a significant concern in Android handheld devices (smartphones, smart glasses, and smartwatch) with limited storage and resources.

(b) The difficulty for the malware writer to understand the obfuscated normal application

To develop malware in certain situations, malware writers need to perform reverse engineering on the normal repackaged application. Therefore it is able to confuse them to steal private information and discover application vulnerabilities from that obfuscated normal @ benign application code (*Diego et al., 2004*).

(c) Security practitioners can detect malware easily

Obfuscation also facilitates the detection of malware by the researcher (*Nissim et al., 2014*). To illustrate, there are certain situations where malware regularly adopts similar obfuscation marks, which is impossible to exist in normal application. Therefore, security practitioners able to detect malware with the presence of these marks. Following all the advantages and drawbacks, continuous research on obfuscation is crucial to obtain better results from the detection of malware through the static analysis.

**Table 10  Studies conducted to overcome obfuscation.** To overcome obfuscation, many studies were conducted on different approaches.

| References | Year | Solution for the obfuscation |
| --- | --- | --- |
| DNADroid (*Crussell, Gibler & Chen, 2012*) | 2012 | Using programme dependence graphs (PDGs), DNADroid can prevent typical program transformations in obfuscation. |
| *Apvrille & Strazzere (2012)* | 2012 | Detects encryption attempts as one of the obfuscation methods |
| DroidAPIMiner (*Aafer, Du & Yin, 2013*) | 2013 | Includes features, which are regularly used for obfuscation, such as substring (), indexOf(), getBytes(), valueOf(), replaceAll(), Append(), getInstance(), doFinal(), and Crypto.spec.DESKeySpec |
| Androsimilar (*Faruki et al., 2013*) | 2013 | Adopts statistically strong file features based on a normalised entropy |
| Droidanalytics (*Zheng, Sun & Lui, 2013*) | 2013 | Only extracts the API calls in methods and classes, which will be executed in the run time. Additionally, the generated signature, which is based on the analyst-defined API, which have the ability to update flexibly. |
| Apposcopy (*Feng et al., 2014*) | 2014 | Includes the predicate inter-component communication (ICC), which allows the writing of signatures, which are resilient to high-level control flow obfuscation. |
| Drebin (*Arp et al., 2014*) | 2014 | DREBIN extracts API calls related to obfuscation and loading of code, such as DexClassLoader.loadClass() and Cipher.getInstance |
| Dendroid (*Suarez-Tangil et al., 2014*) | 2014 | Concentrates on the internal structure of code units (methods) to resist obfuscation. |
| Droidlegacy (*Deshotels, Notani & Lakhotia, 2014b*) | 2014 | Graph node represents the Java class, which detects light obfuscation. |
| *Firdaus (2017)* | 2017 | Uses Jadx (a reverse engineering tool), which provides the de-obfuscation option. It is capable of de-obfuscating the obfuscation code in minimal error. |

### The list of all articles in the detection of malware in static analysis

To identify the trends in the detection of malware through static analysis, this section presents a list of previous works of research, which cover all areas (year, features, and classification). Table 11 lists a study DroidARA (*Fan et al., 2019*) in 2019, which performed an experiment combined with DL and graph and differentiation between malware and normal application. It applied a call graph to extract the API features and convolutional neural network (CNN) for classification. At the time of writing this paper, this is a new trend in detecting Android malware. Therefore, in future research, it is possible to witness more research combination similar to this with different features.

From the lists, most of researchers used API and manifest file features in their experiments to detect the malware. It proofs that API features were the popular codes used by the malware developers to create the malware. The program is based on the Android application package file in the (.apk) format, which is also used to install an application in android-based mobile devices. Every app project must have an androidmanifest.xml file at the root of the project source set. This manifest file is regularly in a binary form inside the APK, however when chosen in the APK Analyzer, the xml form is restructured and produced. There are some

**Table 11  The detection of malware, which attacks Android OS, based on previous static analysis.** To identify the trends in the detection of malware through static analysis, this section presents a list of previous works of research, which cover all areas (year, features, and classification).

| Year | Ref | Features (●) | | | | | | | | | Classification (✓) | | | | |
|------|-----|----|-----|---|----|---|---|---|---|---|----|----|----|-------|-------|
| | | AD | API | A | DP | C | F | G | M | N | CB | ML | DL | Graph | Other |
| 2019 | *Hammad, Bagheri & Malek (2019)* | | | | | | | | ● | | | | | | Multiple-Domain Matrix (MDM) |
| 2019 | *Fan et al. (2019)* | | ● | | | | | | | | | | ✓ | ✓ | |
| 2019 | *Badhani & Muttoo (2019)* | | ● | | | | | | ● | | | ✓ | | | |
| 2019 | *Wu et al. (2019)* | | ● | | | | | | | | | | | ✓ | |
| 2019 | *Liu et al. (2019)* | | ● | | | | | | ● | | | | | ✓ | |
| 2019 | *Alotaibi (2019)* | | ● | | | | | | ● | ● | | ✓ | | | |
| 2019 | *Zhang, Tian & Duan (2019)* | | | | | | | | | | ● | | | ✓ | |
| 2019 | *Alsoghyer & Almomani (2019)* | | ● | | | | | | | | | ✓ | | | |
| 2019 | *Yildiz & Doğru (2019)* | | ● | | | | | | ● | | | ✓ | | | |
| 2019 | *Singh, Jaidhar & Kumara (2019)* | | ● | | | | | | ● | | | ✓ | | | |
| 2019 | *Lei et al. (2019)* | | ● | | | | | | | | | ✓ | | ✓ | |
| 2018 | *Firdaus et al. (2018)* | | | | ● | ● | | | | | ● | ✓ | | | |
| 2018 | *Zhao et al. (2019)* | | | | | | | | | | ● | ✓ | | | |
| 2018 | *Gu et al. (2018)* | | | | | | | | | | ● | | | ✓ | |

**Table 11** (*continued*)

| Year | Ref | Features (●) | | | | | | | | | Classification (✓) | | | | |
|------|-----|:--:|:--:|:--:|:--:|:--:|:--:|:--:|:--:|:--:|:--:|:--:|:--:|:--:|:--:|
| | | AD | API | A | DP | C | F | G | M | N | CB | ML | DL | Graph | Other |
| 2018 | *Firdaus et al. (2017)* | | ● | | ● | | | | ● | | | ✓ | | | |
| 2018 | *Firdaus et al. (2018)* | | | | ● | ● | | | ● | | ● | | | | |
| 2018 | *Klieber et al. (2018)* | | | | | | | | ● | | | | | ✓ | |
| 2018 | *Kumar, Kuppusamy & Aghila (2018)* | | | | | | | | ● | | | ✓ | | | |
| 2018 | *Wang et al. (2018)* | | ● | | | | ● | | ● | | ● | ✓ | | | |
| 2018 | *Ma Zhao-hui et al. (2017)* | | | | | | | | ● | | | ✓ | | | |
| 2017 | *Feizollah et al. (2017)* | | | | | | | | ● | | | ✓ | | | |
| 2017 | *Kim et al. (2018)* | | | | | | | | | | ● | ✓ | | | |
| 2017 | *Zhou et al. (2017)* | | ● | | | | | | | | | | | ✓ | |
| 2017 | *Pooryousef & Amini (2017)* | | | | | | | | ● | | | | | ✓ | |
| 2017 | *Wu et al. (2018)* | | | | | | | | ● | | | | | ✓ | |
| 2017 | *Chang & De Wang (2017)* | | ● | | | | | | ● | | | ✓ | | | |
| 2016 | *Junaid, Liu & Kung (2016a)* | | ● | | | | | | | | | | | ✓ | |
| 2016 | *Atici, Sagiroglu & Dogru (2016)* | | | | | | | | | | ● | ✓ | | ✓ | |
| 2016 | *Medvet & Mercaldo (2016)* | | | | | | | | | | ● | ✓ | | | |
**Table 11** (*continued*)

| Year | Ref | Features (●) | | | | | | | | | | Classification (✓) | | | |
|------|-----|----|-----|---|----|---|---|---|---|---|----|----|----|-------|-------|
| | | AD | API | A | DP | C | F | G | M | N | CB | ML | DL | Graph | Other |
| 2016 | *Wu et al. (2016)* | | ● | | | | | | | | | ✓ | | | |
| 2016 | *Nissim et al. (2016)* | | | | | | | | ● | | | ✓ | | | |
| 2015 | *Sheen, Anitha & Natarajan (2015)* | | ● | | | | | | ● | | | ✓ | | | |
| 2015 | *Kang et al. (2015)* | | ● | | | ● | | | ● | | | ✓ | | | |
| 2015 | *Kang et al. (2015)* | | | ● | | ● | | | ● | | | ✓ | | | |
| 2015 | *Firdaus & Anuar (2015)* | | | | ● | ● | | | | | ● | ✓ | | | |
| 2015 | *Talha, Alper & Aydin (2015)* | | | | | | | | ● | | | ✓ | | | |
| 2015 | *Junaid, Liu & Kung (2016a)* | | ● | | | | | | ● | | | | | ✓ | |
| 2015 | *Lee, Lee & Lee (2015)* | | | | | | | | | | ● | | | | |
| 2015 | *Elish et al. (2015)* | | ● | | | | | | ● | | | | | ✓ | |
| 2015 | *Gordon et al. (2015)* | | ● | | | | | | | | | | | ✓ | |
| 2014 | *Yerima, Sezer & McWilliams (2014)* | | | | | | | | ● | | ● | ✓ | | | |
| 2014 | *Arp et al. (2014)* | | ● | | | | | | ● | ● | | ✓ | | | |
| 2014 | *Deshotels, Notani & Lakhotia (2014b)* | | ● | | | | | | | | | | | ✓ | |

**Table 11** (*continued*)

| Year | Ref | Features (●) | | | | | | | | | Classification (✓) | | | | |
|------|-----|----|-----|---|----|---|---|---|---|---|----|-----|----|----|-------|-------|
| | | AD | API | A | DP | C | F | G | M | N | CB | ML | DL | Graph | Other |
| 2014 | *Feng et al. (2014)* | | ● | | | | | | | | | | | ✓ | |
| 2014 | *Huang et al. (2014a)* | | ● | | | | | | ● | | | | | ✓ | |
| 2014 | *Steven Arzt et al. (2014)* | | | | | | | | ● | | | | | ✓ | |
| 2014 | *Yerima, Sezer & Muttik (2014)* | | ● | | | ● | | | ● | | | ✓ | | | |
| 2014 | *Seo et al. (2014)* | | ● | | | | | | | | ● | | | | MD5 signature |
| 2014 | *Suarez-Tangil et al. (2014)* | | | | | | | | | | ● | | | ✓ | |
| 2013 | *Aafer, Du & Yin (2013)* | | ● | | | | | | | | | ✓ | | | |
| 2013 | *Lee & Jin (2013)* | | ● | | | | | | | | | | | | |
| 2013 | *Shuying Liang et al. (2013)* | | ● | | | | | | ● | | | | | ✓ | |
| 2013 | *Zhou et al. (2013)* | | ● | | | | | | ● | | | | | ✓ | |
| 2013 | *Luoshi et al. (2013)* | | ● | | | | | | ● | ● | | | | ✓ | |
| 2013 | *Peiravian & Zhu (2013)* | | ● | | | | | | ● | | | ✓ | | | |
| 2013 | *Samra, Kangbin & Ghanem (2013)* | | | ● | | | | | ● | | | ✓ | | | |

**Table 11** (*continued*)

| Year | Ref | Features (●) | | | | | | | | | Classification (✓) | | | |
|------|-----|----|-----|---|----|---|---|---|---|---|----|----|----|-------|-------|
| | | AD | API | A | DP | C | F | G | M | N | CB | ML | DL | Graph | Other |
| 2013 | *Gascon et al. (2013)* | | | | | | | | | | ● | ✓ | | | |
| 2013 | *Huang, Tsai & Hsu (2012)* | | | | | | | | ● | | | ✓ | | | |
| 2013 | *Aung & Zaw (2013)* | | | | | | | | ● | | | ✓ | | | |
| 2013 | *Faruki et al. (2013)* | | | | | | | | | | ● | | | | Similarity digest hashing |
| 2013 | *Paturi et al. (2013)* | | | | | | | | | | ● | | | | Normalised Compression Distance (NCD) |
| 2013 | *Apvrille & Apvrille (2013)* | | | | | | | | | | ● | ✓ | | | |
| 2013 | *Yerima et al. (2013)* | | | | | | | | ● | | ● | ✓ | | | |
| 2013 | *Borja Sanz et al. (2013)* | | | | | | | | ● | | | ✓ | | | |
| 2012 | *Grace et al. (2012a)* | ● | | | | | | | | | | | | ✓ | |
| 2012 | *Wu et al. (2012)* | | ● | | | | | | ● | | | ✓ | | | |
| 2012 | *Bartel et al. (2012)* | | ● | | | | | | ● | | | | | ✓ | |
| 2012 | *Grace et al. (2012b)* | | ● | | | | | | ● | | | | | ✓ | |
| 2012 | *Zhou et al. (2012)* | | | ● | | | | | | | | | | | Fuzzy hashing technique |
| 2012 | *Hao Peng et al. (2012)* | | | | | | | | ● | | | ✓ | | | |

| Year | Ref | Features (●) | | | | | | | | | Classification (✓) | | | | |
|---|---|---|---|---|---|---|---|---|---|---|---|---|---|---|---|
| | | AD | API | A | DP | C | F | G | M | N | CB | ML | DL | Graph | Other |
| 2012 | *Walenstein, Deshotels & Lakhotia (2012)* | | | | | | | | | | ● | | | ✓ | |
| 2012 | *Sahs & Khan (2012)* | | | | | | | | ● | | | ✓ | | ✓ | |
| 2012 | *Sanz et al. (2013)* | | | | | | | | ● | | | ✓ | | | |
| 2012 | *Apvrille & Strazzere (2012)* | | ● | | | | | ● | ● | | | | | | Risk score weight |
| 2012 | *Lu et al. (2012)* | | | | | | | | ● | | | | | ✓ | |
| 2012 | *Crussell, Gibler & Chen (2012)* | | | | | | | | | | ● | | | ✓ | |
| 2012 | *Desnos (2012b)* | | | | | | | | | | ● | | | | Normalised Compression Distance (NCD) |
| 2012 | *Sarma et al. (2012)* | | | | | | | | ● | | | ✓ | | | |
| 2012 | *Yang & Yang (2012)* | | ● | | | | | | ● | | | | | ✓ | |
| 2011 | *Grace et al. (2011)* | | | | | | | | ● | | ● | | | ✓ | |
| 2010 | *Shabtai, Fledel & Elovici (2010)* | | | ● | | | | | ● | | ● | ✓ | | | |
| 2009 | *Aubrey-Derrick Schmidt et al. (2009a)* | | | | | | ● | | | | | ✓ | | | |

**Notes.**

AD, Advertisement libraries; API, API; A, apk, dex and XML properties; DP, Directory path; C, Commands; F, unction call; G, Geographic; M, Manifest file; N, Network address or URLs; CB, Code-based.

changes of the androidmanifest.xml document from a library application depends on was converged into the last androidmanifest.xml record. Other package files fall down into apk, xml and dex properties feature.

Besides the combination of DL and graph, ML and graph were also combined in the studies by *Atici, Sagiroglu & Dogru (2016)* in 2016 and *Sahs & Khan (2012)* in 2012. These studies utilised a similar graph, which was the Control flow graph (CFG), indicating that the combination of ML and graph increased the detection result. Therefore, future work is suggested to test this combination in different static features. Other parts of classification (Multiple-Domain Matrix (MDM), MD5 signature, similarity digest hashing, normalized compression distance (NCD), and fuzzy hashing technique) were also useful in the detection of malware with static features. These classifications also contributed to the availability of future work combinations with ML, DL, and graph.

### Detect unknown malware

Initially, static analysis is unable to discover new or unknown malware as it only examined the code of the application without executing it. To elaborate on this situation, certain malware only executes certain parts whenever the application runs. Provided the drawback of static analysis in the identification of unknown malware, many security practitioners started to adopt machine learning, such as (*Lee, Lee & Lee, 2015*; *Yerima, Sezer & McWilliams, 2014*), Drebin (*Arp et al., 2014*; *Yerima, Sezer & Muttik, 2014*), Droidapiminer (*Aafer, Du & Yin, 2013*; *Apvrille & Apvrille, 2013*), Androsimilar (*Faruki et al., 2013*; *Lee & Jin, 2013*; *Yerima et al., 2013*; *Paturi et al., 2013*; *Shabtai, Fledel & Elovici, 2010*), and (*Firdaus & Anuar, 2015*). Similarly, the graph approach was also a suitable approach for this identification, as shown in *Elish et al. (2015)*, Riskranker (*Grace et al., 2011*), and Dendroid (*Suarez-Tangil et al., 2014*). The *Elish et al. (2015)* study utilised a data dependence graph (DDG), Riskranker, and Dendroid, which employed the control-flow graph (CFG).

### Combination of static and dynamic analyses

It was proven in *Moser, Kruegel & Kirda (2007)* that static analysis was inadequate for the detection of malware as this analysis should be combined with dynamic analysis to detect the malware effectively. Compared to static analysis, the dynamic analysis can evade the obfuscation technique. Essentially, provided that each type of analysis (static and dynamic) has its advantages and drawbacks, the combination of static and dynamic analyses would increase the effectiveness of the countermeasure action on the malware.

### Resource consumption in Android OS and deep learning

Deep learning (DL) is a subset of machine learning in artificial intelligence (AI), which is also known as a deep neural network or deep neural learning. Notably, with unlabeled and unstructured data, DL is capable of learning and predicting the output. It imitates the human brain in data processing, development of patterns from that data, and the implementation of decision making. It could be seen from the current trends that deep learning (DL) technique has a potential for further discovery. The implementation of this technique enables the DL to automatically determine the ideal features for prediction

and classification. Currently, DL is widely used in almost every area, such as large scale image recognition tasks, automated driving, new cancer cell detection, hearing and speech translation, and aerospace area identification (*Mathworks, 2020*).

However, DL requires substantial computing power, which needs a high amount of the graphic processing unit (GPU) based on the data to be processed (*Mathworks, 2020*). This situation leads to an issue in the detection of malware, which attacks Android devices. Provided that Android mobile device is a device with small computing power, the adoption of DL becomes the main concern. However, the transfer of information from the Android device to the cloud provider is possible only for the execution of the DL process, which would then develop the device. Therefore, the large-scale adoption of DL is possible for future work in the static analysis.

### Future work in static analysis

It could be seen from the review in the previous sections ('Survey Methodology') that many future opportunities for the static analysis to detect the malware, which attacks the Android. One of the opportunities is the combination of different DL classifier (Recurrent neural network—RNN, Generative* adversarial networks—GAN or Deep belief network*—DBN) with other features besides API, with different types of graph. However, 'The list of all articles in the detection of malware in static analysis' shows that only one experiment started the combination between DL and graph with one API feature in 2019. Therefore, the accuracy of detection results would increase, leading to the identification of a new family of malware. It is also noteworthy that other future alternatives are available for the combination of DL with NCD and MDM.

### Popular features (API and manifest file)

'The list of all articles in the detection of malware in static analysis' shows that many static analysis researchers frequently applied the manifest file and API calls as the features in their experiments. To illustrate, these popular features had been examined by the researchers from 2010 until 2019 due to the official update for Android and the addition of new features from time to time. However, most of the malwares still utilised similar features within this timespan, while other malwares utilised the latest and updated features. Therefore, the researchers are required to place continuous focus on these popular features.

The manifest file is one file with numerous features in it, such as permission, intent, hardware component, and application component, while API is a ready code for the programmer to develop their application. Therefore, it is crucial for researchers to thoroughly scrutinise these two categories of features. Moreover, some researchers incorporated other features to combine the manifest and API for more effective malware detection.

## CONCLUSIONS

Following the interest to explore the recent studies in the static analysis, a review was performed on the existing studies by past security investigators on Android malware detection, which was explained through phases (reverse engineer, features, and

classification). Furthermore, this review covered the information within the ten years range (2009 to 2019). In this article, the features used in the static analysis were also reviewed. Within the aforementioned timespan, many security practitioners still preferred the API and manifest files, indicating the relevance of these two features. Moreover, the latest trends in classification were highlighted, which consists of machine learning, deep learning, graph, and other methods. These trends have proven the relevance of the graph method compared to machine learning. Static analysis researchers began the adoption of deep learning in their detection. This article also discussed the open research issues in the static analysis, including obfuscation as one of the weaknesses of static analysis and the methods of overcoming it. Many static analysis researchers implemented multiple methods to solve obfuscation and achieve a successful malware detection, such as concentrating the codes with the implementation of obfuscation (DexClassLoader.loadClass, Crypto.spec.DESKeySpec, and Cipher.getInstance) using a tool with the de-obfuscation option, including the adoption of graph node and program dependence graphs (PDGs).

### Funding
This work was supported by the Ministry of Higher Education (MOHE) for Fundamental Research Grant Scheme (FRGS) with grant number RDU190190, FRGS/1/2018/ICT02/UMP/02/13, and Universiti Malaysia Pahang (UMP) internal grant with grant number RDU1803142. The funders had no role in study design, data collection and analysis, decision to publish, or preparation of the manuscript.

### Grant Disclosures
The following grant information was disclosed by the authors:
Ministry of Higher Education (MOHE).
Fundamental Research Grant Scheme (FRGS): RDU190190, FRGS/1/2018/ICT02/UMP/02/13.
Universiti Malaysia Pahang (UMP): RDU1803142.

### Competing Interests
The authors declare there are no competing interests.

### Author Contributions
- Rosmalissa Jusoh conceived and designed the experiments, performed the computation work, prepared figures and/or tables, and approved the final draft.
- Ahmad Firdaus conceived and designed the experiments, performed the computation work, authored or reviewed drafts of the paper, and approved the final draft.
- Shahid Anwar and Mohd Zamri Osman performed the experiments, authored or reviewed drafts of the paper, and approved the final draft.
- Mohd Faaizie Darmawan analyzed the data, prepared figures and/or tables, and approved the final draft.

- Mohd Faizal Ab Razak analyzed the data, prepared figures and/or tables, and approved the final draft.

## Data Availability

This is a review article.

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
