# Peer review of "Malware detection using static analysis in Android: a review of FeCO (features, classification, and obfuscation)"

_PeerJ Computer Science, doi:10.7717/peerj-cs.522_

## Round 0.1 · original submission · Major Revisions

Since this is a survey paper, more details are needed. The reviewers pointed out that reviewing high impact journals (for instance IEEE, Springer, etc.) is necessary.

Reviewer 1 ·

Basic reporting

Thanks to the author for using a clear and understandable English in article. It's nice to mention dynamic analysis and static analysis in the article. however, there are a few more approaches in the literature. For example, these are signature-based approach and hybrid approach. The author should mention them as well. In my opinion, the author can add them to line 61. The author should briefly explain these approaches in the article.

Between lines 63 and 71, a comparison of Dynamic and Static analysis has been tried. It would be more appropriate to include this in a comparison table with citations. The author can add the signature-based approach and hybrid approaches in this table.

As the author mentioned in line 119, He used “static analysis”, “malware”, and “Android” in the CA database for his literature review. however, the keywords of the article are; Android; Review; Static analysis; Machine learning; Features. Author should add “malware” here like as others.

Table 2 and 7, Figure 9 can be removed. The author should explain “Figure 5” in section 2.3. Feature more. A little more explanation for Figure 5 in there is better.

A comparison table with 4 machine learning (CL, RL, NN&DL and EM) should be created with citations. This table should show the machine learning metrics (such as KNN,SVM,K Means, Deep Q,MLP or RF) used as well. This table created can be added to section 2.4.1.This section should explain the advantages and disadvantages of compared machine learning methods. The advantages and disadvantages of static analysis against these ML methods should also be mentioned.

Table 9 mentions “studies conducted to overcome obfuscation”. Open source advantages and disadvantages in section 3.1 can also be added to this table with citations.

Table 10 is very nice. It is listed by years. The 2nd listed criterion may be reference numbers. The reference numbers in the table can be listed in ascending order. So it can look better

The interpretation of table 10 should be given more in section 3.3. It is useful to add explanations such as which ML was used the most and why it was preferred. The frequency of using static and dynamic analysis in studies can also be mentioned. Is the data set used or not used in the studies? Similar interpretations should be made for table

Check whether the web links in references are accessible. The author can use ”webcitation” for such links.

Experimental design

no comment

Validity of the findings

no comment

Additional comments

All section numbers in the article should be rechecked. It seems to me that it will be “3.2.1. Research to counter obfuscation” instead of 4.2.1. and it should be 3.2.2. instead of 4.2.2 as well.

Reviewer 2 ·

Basic reporting

In this study, a review has been made on malware detection systems in the Android operating system, which is one of the current topics in the literature. It can be important as current issues are covered in the study. However, serious deficiencies are observed in the writing style and organization of this study. In addition, the studies examined need to be detailed. The study seems superficial in this state. For this reason, I see benefit in reorganizing the study. Otherwise, the work will look too disorganized and difficult to understand.

Experimental design

My suggestions are as follows:
Which journals have articles listed most at the end of your research? In other words, which journals have been used more? Survey publications are especially important for new researchers who will work in this field. How many articles from the most frequently used 5 or 10? These journals should be given in a table. In this way, it will be revealed which journals will mostly benefit people who will conduct research in this field. These journals can be added by opening a new subsection in Section 1 or where appropriate.
When referencing tables and figures, there is no need to use the above or below patterns. It is already clear which table or figure it is. As a matter of fact, the tables or figures in the munuscript I read are in a different file. Instead of the above or below, a statement like in Table x, Figure y will be more fluent.
The data sets used in this field are given in Section 2.1. When I examine some of the references provided in this section, I only see the names of the relevant data sets in the reference list. For example, in [31, 32, 34, 35] references only the names of the relevant data sets are written. The official link of the data set should be added to these references. If there are blank references other than these, they should also contain the necessary information. This is important for the reader.
While "apktool" and "aapt" tools are expressed separately in Figure 4, these two tools are given as "Apktool @ aapt" in Table 5. Why is it included in Table 5? Is it given as “Apktool @ aapt” because the related works use both tools? I suggest it to be given separately in the works using the "apktool" and "aapt" tools. If it will be used together, the necessary explanation should be made. Otherwise, the reader will be confused.
Are there any studies using the Reinforcement learning technique under section 2.4.1? While the studies using other techniques are given in Table form, there is no table related to this technique. If there are no studies using this technique, an explanation should be made about it.
Some paragraphs consist of one or two sentences. Paragraphs should be expanded using meaningful sentences or combined with other paragraphs in a way that does not disturb the integrity of the part.
On line 562, a section titled "Open research issues" is written. What is explained with the chapter title is not compatible. When I read the title, I thought it would give obvious problems in this area, but I did not encounter such a thing. I was disappointed as a result. This section needs to be rewritten.
There is a sentence in the subsection on line 637. This subsection should be expanded in line with the relevant tables.
When Table 10 is examined, A and M are similar to each other. This needs to be parsed. Because "Manifest file" represented by M can be evaluated as a subset of "apk, dex and XML properties" represented by A. This will make it difficult for the reader to understand.

Validity of the findings

Reviewed articles in this study should be detailed.

Additional comments

In this study, a review has been made on malware detection systems in the Android operating system, which is one of the current topics in the literature. It can be important as current issues are covered in the study. However, serious deficiencies are observed in the writing style and organization of this study. In addition, the studies examined need to be detailed. The study seems superficial in this state. For this reason, I see benefit in reorganizing the study. Otherwise, the work will look too disorganized and difficult to understand.

---

## Round 0.2 · Major Revisions

The paper needs detailed proofreading. The references should be checked. Moreover, important and recent references are missing in malware detection.

Reviewer 2 ·

Basic reporting

A comprehensive survey study has emerged. I think contribution has been made to the field of Android malware detection.

Experimental design

There is no problem.

Validity of the findings

There is no problem.

Additional comments

The authors made the suggested changes. The study is acceptable.

Reviewer 3 ·

Basic reporting

Survey methodology and research methods are well explained. The Introduction section provides sufficient information about the general content of the study.

Experimental design

1- Different reverse engineering tools are mentioned in Section 2.2. The studies they are used and the way they are used are expressed. However, the reverse engineering tools themselves are not cited.

2- In the description of Figure 8, the expression "(B) However, DL involves automatic feature selection and malware classification" is used. However, in the B part of the figure, "feature extraction + classification" expression is used. This situation causes the two different terms feature extraction and feature selection to be perceived as the same thing. Likewise, there is the same confusion in the paragraph starting at line 470.

3- At the end of the paragraph starting on line 470, publication 106 is cited. However, I could not see the direct relationship of the cited work with DL or automatic feature extraction. In Table 8, publication 106 is cited again and the year of publication is written as 2015. However, the year of publication 106 in the references is 2019. I think there is a confusion in the citations.

4- Although the last reference number is 138 in the references section, I see that there are higher numbered references in Table 11. I think there are missing references in the references table.

Validity of the findings

no comment

Additional comments

The research methods and organization of the study are very good. It clearly presents the boundaries of the researched area to the reader. However, the confusion, especially in references, needs to be corrected.

---

## Round 0.3 · accepted · Accept

According to the reviewers' comments, the paper is ready for publication after minor edits. We kindly ask the authors to check the paper for typos and language corrections.

Reviewer 1 ·

Basic reporting

the article meets the standards, with suggested improvements.

Experimental design

issues given for previous version are completed.

Validity of the findings

Conclusions are well presented, related to research question & limited to supporting results.

Reviewer 2 ·

Basic reporting

Some references do not show the year, volume, journal name. Missing references need to be edited.

Experimental design

No comment

Validity of the findings

No comment

Additional comments

1) In Table 3, the journal "Expert Systems with Applications" is written twice. You can delete one of them.
2) A dot was used before the reference in lines 265 and 266. This should be fixed. “Feature selection is important in order to increase the accuracy of the detection system. [18], [42], [43]. " -> "Feature selection is important in order to increase the accuracy of the detection system [18], [42], [43]."
3) The concepts of feature extraction and feature selection are mixed. It is not the feature selection described in the subsection 2.3, but the feature extraction. Obtaining attributes such as permissions and APIs from application files is feature extraction. Attribute selection is completely different. This must be edited.

Reviewer 3 ·

Basic reporting

no comment

Experimental design

no comment

Validity of the findings

no comment

Additional comments

The authors made the suggested changes.. In my opinion, the manuscript is acceptable.